# Imprecise Bayesian Neural Networks

## Abstract

Uncertainty quantification and robustness to distribution shifts are important goals in machine learning and artificial intelligence. Although Bayesian Neural Networks (BNNs) allow for uncertainty in the predictions to be assessed, different sources of uncertainty are indistinguishable. We present Imprecise Bayesian Neural Networks (IBNNs); they generalize and overcome some of the drawbacks of standard BNNs. These latter are trained using a single prior and likelihood distributions, whereas IBNNs are trained using credal prior and likelihood sets. They allow to distinguish between aleatoric and epistemic uncertainties, and to quantify them. In addition, IBNNs are more robust than BNNs to prior and likelihood misspecification, and to distribution shift. They can also be used to compute sets of outcomes that enjoy probabilistic guarantees. We apply IBNNs to two case studies. One, for motion prediction in autonomous driving scenarios, and two, to model blood glucose and insulin dynamics for artificial pancreas control. We show that IBNNs performs better when compared to an ensemble of BNNs benchmark.

## 1 Introduction

One of the greatest virtues an individual can have is arguably being aware of their own ignorance, and acting cautiously as a consequence. Similarly, an autonomous system using neural networks (NNs) would greatly benefit from understanding the probabilistic properties of the NN's output (e.g., variance, robustness to distribution shift), in order to incorporate them into any further decision-making. In this paper, we present a generalization of Bayesian neural networks that allows us to give a machine such a desirable quality.

In the last few years, there has been a proliferation of work on calibrating (classification) NNs, in order to estimate the confidence in their outputs (Guo et al., 2017) or to produce conformal sets that are guaranteed to contain the true label, in a probably approximately correct (PAC) sense (Park et al., 2020). While such methods are a promising first step, they require a calibration set (in addition to the original training set) and cannot be directly used on out-of-distribution data without further examples.

Bayesian neural networks (BNNs) offer one approach to overcome the above limitations. The Bayesian paradigm provides a rigorous framework to analyze and train uncertainty-aware neural networks, and more generally to support the development of learning algorithms (Jospin et al., 2022). In addition, it overcomes some of the drawbacks of deep learning models, namely that they are prone to overfitting, which adversely affects their generalization capabilities, and that they tend to be overconfident about their predictions when they provide a confidence interval. BNNs, though, are trained using a single prior, which may still suffer from miscalibration and robustness issues (Lenk & Orme, 2009).

In this work we introduce imprecise Bayesian neural networks (IBNNs). Unlike other techniques in the fields of artificial intelligence (AI) and machine learning (ML) involving imprecise probabilities – that typically only focus on classification problems – IBNNs can be used for classification, prediction, and regression. They capture the ambiguity the designer faces when selecting which prior to choose for the parameters of a neural network and which likelihood distribution to choose for the training data at hand. An IBNN can be defined as a NN trained using credal prior and likelihood sets. Credal sets are convex sets of probability measures (see Remark 1); we use them to train IBNNs in order to overcome some of the drawbacks of BNNs. In particular, they allow to counter the criticism to the practice in (standard) Bayesian statistics of (i) using

a single, arbitrary prior to represent the initial state of ignorance of the agent, (ii) using non-informative priors to model ignorance, and (iii) using a single, arbitrary likelihood to represent the agent's knowledge about the sampling model.[1] As a consequence, credal sets make the analysis more robust to prior and likelihood misspecification. In addition, they make it possible to quantify and distinguish between epistemic (i.e. reducible) and aleatoric (i.e. irreducible) uncertainties. This is desirable in light of several areas of recent ML research, such as Bayesian deep learning (Depeweg et al., 2018; Kendall & Gal, 2017), adversarial example detection (Smith & Gal, 2018), and data augmentation in Bayesian classification (Kapoor et al., 2022).

The motivation for training an artificial NN using credal sets is twofold: (i) a single probability distribution does not suffice to represent ignorance in the sense of lack of knowledge (Hüllermeier & Waegeman, 2021), and (ii) working with credal sets allows to be robust against prior and likelihood misspecification. A more in-depth discussion can be found in Appendix A. In addition, we point out that despite a hierarchical Bayesian model (HBM) approach seems to be a viable alternative to one based on credal sets, these latter are better justified philosophically, and do not suffer from the same theoretical shortcomings of HBM approaches (Bernardo, 1979; Hüllermeier & Waegeman, 2021; Jeffreys, 1946; Walley, 1991). A more detailed explanation can be found in Appendix B.

We summarize our contributions next: (1) We present IBNNs, and develop the theoretical tools and the procedure required to use them in practice. (2) We present theoretical results to show that IBNNs are more robust than BNNs to prior and likelihood misspecification, and to distribution shifts. We also prove how IBNNs can be used to specify sets of outcomes that enjoy probabilistic guarantees. (3) We apply IBNNs to model two safety critical systems. One, motion prediction for autonomous driving, and two, the human insulin and blood glucose dynamics for artificial pancreas control. We demonstrate improvements in both these settings with respect to ensemble of BNNs methods.

Before moving on, let us point out that while IBNNs pay a computational price coming from the use of credal sets, they are able to quantify epistemic uncertainty, unlike BNNs, and to do so in a principled manner, unlike ensemble of BNNs. In addition, they require less stringent assumptions on the nature of the prior and likelihood ambiguity faced by the agent than other imprecise-probabilities-based techniques, as explained in Appendices B and M. We also stress that if the scholar prioritizes computational efficiency, they should use backpropagation-based methods, as they are much faster than Bayesian techniques. In safety-critical situations instead – where using uncertainty-informed methods is crucial for the downstream task performance – IBNNs are the natural choice.

**Structure of the paper.** Section 2 presents the needed preliminary concepts, followed by section 3 that introduces some important theoretical results. We discuss the applied aspects of IBNNs in in section 4. We present our experimental results in section 5, and we examine the related work in section 6. Section 7 concludes our work. In the appendices, we give further theoretical and philosophical arguments and we prove our claims.

## 2 Background and Preliminaries

In this section, we present the background notions that are needed to understand our main results. In section 2.1 we introduce Bayesian neural networks, and section 2.2 discusses (finitely generated) credal sets, upper probabilities and lower probabilities. The reader that is familiar with these concepts can skip to section 3.

### 2.1 Bayesian neural networks

In line with the recent survey on BNNs by Jospin et al. (2022), Bayes' theorem can be stated as $P(H \mid D) = [P(D \mid H)P(H)]/P(D) = P(D, H)/ \int P(D, H')\mathrm{d}H'$, where $H$ is a hypothesis about which the agent holds some prior beliefs, and $D$ is the data the agent uses to update their initial opinion. Probability distribution $P(D \mid H)$ represents how likely it is to observe data $D$ if hypothesis $H$ were to be true, and is called *likelihood*, while probability distribution $P(H)$ represents the agent's initial opinion around the plausibility

---

[1]Criticisms (i) and (iii) are also pointed out in Manchingal & Cuzzolin (2022, Section 2.2).

of hypothesis $H$, and is called *prior*. The *evidence* available is encoded in $P(D) = \int P(D, H')\mathrm{d}H'$, while *posterior* probability $P(H \mid D)$ represents the agent's updated opinion. Using Bayes' theorem to train a predictor can be understood as learning from data $D$: the Bayesian paradigm offers an established way of quantifying uncertainty in deep learning models.

BNNs are stochastic artificial neural networks (ANNs) trained using a Bayesian approach (Goan & Fookes, 2020; Jospin et al., 2022; Lampinen & Vehtari, 2001; Titterington, 2004; Wang & Yeung, 2021). The goal of ANNs is to represent an arbitrary function $y = \Phi(x)$. Let $\theta$ represent the parameters of the network, and call $\Theta$ the space $\theta$ belongs to. Stochastic neural networks are a type of ANN built by introducing stochastic components to the network. This is achieved by giving the network either a stochastic activation or stochastic weights to simulate multiple possible models with their associated probability distribution. This can be summarized as $\theta \sim p(\theta)$, $y = \Phi_\theta(x) + \varepsilon$, where $\Phi$ depends on $\theta$ to highlight the stochastic nature of the neural network, $p$ is the density of a probability measure $P$ on $\Theta$,[2] and $\varepsilon$ represents random noise to account for the fact that function $\Phi_\theta$ is just an approximation.

To design a BNN, the first step is to choose a deep neural network *architecture*, that is, functional model $\Phi_\theta$. Then, the agent specifies the *stochastic model*, that is, a prior distribution over the possible model parametrization $p(\theta)$, and a prior confidence in the predictive power of the model $p(y \mid x, \theta)$. Given the usual assumption that multiple data points from the training set are independent, the product $\prod_{(x,y) \in D} p(y \mid x, \theta)$ represents the *likelihood* of outputs $y \in D_\mathbf{y}$ given inputs $x \in D_\mathbf{x}$ and parameter $\theta$, where (a) $D = D_\mathbf{x} \times D_\mathbf{y}$ is the training set; (b) $D_\mathbf{x} = \{x_i\}_{i=1}^n$ is the collection of training inputs, which is a subset of the space $\mathcal{X}$ of inputs; (c) $D_\mathbf{y} = \{y_i\}_{i=1}^n$ is the collection of training outputs, which is a subset of the space $\mathcal{Y}$ of outputs. In the paper, we call "likelihood" both $p(y \mid x, \theta)$ and $\prod_{(x,y) \in D} p(y \mid x, \theta)$, as no confusion arises.

The model parametrization can be considered to be hypothesis $H$. Following Jospin et al. (2022), we assume independence between model parameters $\theta$ and training inputs $D_\mathbf{x}$, in formulas $D_\mathbf{x} \perp\!\!\!\perp \theta$. Hence, Bayes' theorem can be rewritten as

$$p(\theta \mid D) = \frac{p(D_\mathbf{y} \mid D_\mathbf{x}, \theta)p(\theta)}{\int_\Theta p(D_\mathbf{y} \mid D_\mathbf{x}, \theta')p(\theta')\mathrm{d}\theta'} \propto p(D_\mathbf{y} \mid D_\mathbf{x}, \theta)p(\theta).$$

Notice that the equality comes from having assumed $D_\mathbf{x} \perp\!\!\!\perp \theta$. Posterior density $p(\theta \mid D)$ is high dimensional and highly nonconvex (Izmailov et al., 2021b; Jospin et al., 2022), so computing it and sampling from it is a difficult task. The first issue is tackled using Variational Inference (VI) procedures, while Markov Chain Monte Carlo (MCMC) methods address the second challenge. Both are reviewed – in the context of machine learning – in Jospin et al. (2022, Section V), where the authors also inspect their limitations. BNNs can be used for prediction, regression, and classification (Jospin et al., 2022, Section II); besides having a solid theoretical justification, there are practical benefits from using BNNs, as presented in Jospin et al. (2022, Section III).

## 2.2 Imprecise probabilities

As IBNNs are rooted in the theory of imprecise probabilities (IPs), in this section we give a gentle introduction to the IP concepts we will use throughout the paper.

IBNNs are based on the *Bayesian sensitivity analysis* (BSA) approach to IPs, that in turn is grounded in the *dogma of ideal precision* (DIP) (Berger, 1984), (Walley, 1991, Section 5.9). The DIP posits that in any problem there is an *ideal probability model* which is precise, but which may not be precisely known. We call this condition *ambiguity* (Ellsberg, 1961; Gilboa & Marinacci, 2013).

Facing ambiguity can be represented mathematically by a set $\mathcal{P}$ of priors and a set $\mathcal{L}$ of likelihoods that seem "plausible" or "fit" to express the agent's beliefs on the parameters of interest and their knowledge of the data generating process (DGP). Generally speaking, the farther apart the "boundary elements" of the sets (i.e. their infimum and supremum), the higher the agent's ambiguity. Of course, if $\mathcal{P}$ and $\mathcal{L}$ are singletons we go back to the usual Bayesian paradigm.

---

[2]We can write $p$ as the Radon-Nikodym derivative of $P$ with respect to some $\sigma$-finite dominating measure $\mu$, that is, $p = \mathrm{d}P/\mathrm{d}\mu$.

A procedure based on sets $\mathcal{P}$ and $\mathcal{L}$ yields results that are more robust to prior and likelihood misspecification than a regular Bayesian method. In the presence of prior ignorance and indecisiveness about the sampling model, it is better to give answers in the form of intervals or sets, rather than arbitrarily select a prior and a likelihood, and then update. Sets $\mathcal{P}$ and $\mathcal{L}$ allow to represent *indecision*, thus leading to less informative but more robust conclusions.

**Remark 1.** *Throughout the paper, we denote by $\Pi = \{P_1, \ldots, P_k\}$, $k \in \mathbb{N}$, a finite set of probabilities on a generic space $\Omega$, such that for all $j \in \{1, \ldots, k\}$, $P_j$ cannot be written as a convex combination of the other $k-1$ components of $\Pi$. We denote by $\Pi'$ its convex hull $\Pi' \equiv Conv\Pi$, i.e., the set of probabilities $Q$ on $\Omega$ that can be written as $Q(A) = \sum_{j=1}^{k} \alpha_j P_j(A)$, for all $A \subset \Omega$, where the $\alpha_j$'s are elements of $[0,1]$ that sum up to 1. In the literature, it is referred to as a finitely generated credal set (Cozman, 2000; Levi, 1980). Notice then that the extreme elements of $\Pi'$ correspond to the elements of $\Pi$, that is, $ex\Pi' = \Pi$. Simple graphical representations of finitely generated credal sets are given in Figures 1 and 2.*

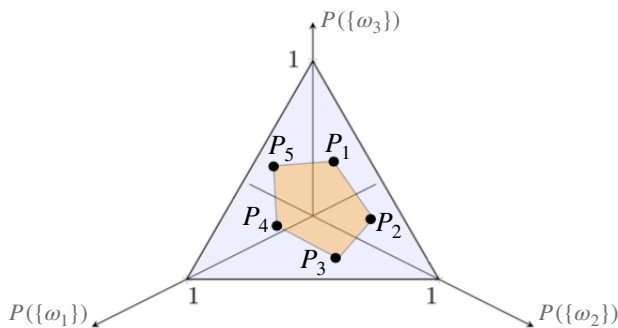

Figure 1: Suppose we are in a 3-class classification setting, so $\Omega = \{\omega_1, \omega_2, \omega_3\}$. Then, any probability measure $P$ on $\Omega$ can be seen as a probability vector. For example, suppose $P(\{\omega_1\}) = 0.6$, $P(\{\omega_2\}) = 0.3$, and $P(\{\omega_3\}) = 0.1$. We have that $P \equiv (0.6, 0.3, 0.1)^\top$. Since its elements are positive and sum up to 1, probability vector $P$ belongs to the unit simplex, the purple triangle in the figure. Then, we can specify $\Pi = \{P_1, \ldots, P_5\}$, and obtain as a consequence that $\Pi' = Conv\Pi$ is the orange pentagon. It is a convex polygon with finitely many extreme elements, and it is the geometric representation of a finitely generated credal set.

Let us now introduce the concepts of *lower* and *upper probabilities*. The lower probability $\underline{P}$ associated with $\Pi$ is given by $\underline{P}(A) = \inf_{P \in \Pi} P(A)$, for all $A \subset \Omega$. The upper probability $\overline{P}$ associated with $\Pi$ is defined as the conjugate to $\underline{P}$, that is, $\overline{P}(A) := 1 - \underline{P}(A^c) = \sup_{P' \in \Pi} P'(A)$, for all $A \subset \Omega$. These definitions hold even if $\Pi$ is not finite. Then, we have the following important result.

**Proposition 2.** *$\overline{P}$ is the upper probability for $\Pi$ if and only if it is also the upper probability for $\Pi'$. That is, $\overline{P}(A) = \sup_{P \in \Pi} P(A) = \sup_{P' \in \Pi'} P'(A)$, for all $A \subset \Omega$.*

A version of Proposition 2 for finitely additive probability measures can be found in Walley (1991, Section 3.6). A simple graphical representation of upper and lower probabilities for a set $A$ is given in Figure 2.

## 3 Theoretical results

In this section, we provide the procedure to follow in order to compute the posterior credal set in the context of IBNNs, and we show how IBNNs are able to capture both aleatoric and epistemic uncertainties associated with the analysis. The training method is presented in Section 3.1. We show that IBNNs are more robust to distribution shifts than regular BNNs as a result of Proposition 4. We first give the formal definition of an IBNN.

**Definition 3** (IBNN). *An IBNN is a stochastic artificial neural network trained using finitely generated credal prior and likelihood sets.*

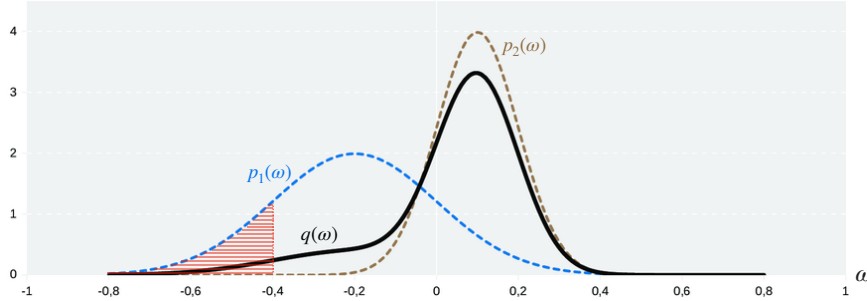

Figure 2: In this figure, a replica of Flint et al. (2017, Figure 1), $\Pi = \{P_1, P_2\}$, where $P_1$ and $P_2$ are two Normal distributions whose pdf's $p_1$ and $p_2$ are given by the dashed blue and brown curves, respectively. Their convex hull is $\Pi' = \mathrm{Conv}\Pi = \{Q : Q = \beta P_1 + (1 - \beta)P_2, \text{ for all } \beta \in [0, 1]\}$. The pdf $q$ of an element $Q$ of $\Pi'$ is depicted by a solid black curve. In addition, let $A = [-0.8, -0.4]$. Then, $\underline{P}(A) = \int_{-0.8}^{-0.4} p_2(\omega)\mathrm{d}\omega \approx 0$, while $\overline{P}(A)$ is given by the red shaded area under $p_1$, that is, $\overline{P}(A) = \int_{-0.8}^{-0.4} p_1(\omega)\mathrm{d}\omega$.

### 3.1 IBNN procedure

Recall that $D = D_{\mathbf{x}} \times D_{\mathbf{y}}$ denotes the training set, where $D_{\mathbf{x}} = \{x_i\}_{i=1}^n \subset \mathcal{X}$ is the collection of training inputs, $D_{\mathbf{y}} = \{y_i\}_{i=1}^n \subset \mathcal{Y}$ is the collection of training outputs, and $\mathcal{X}$ and $\mathcal{Y}$ denote the input and output spaces, respectively. We then denote by $P$ a generic prior on the parameters $\theta \in \Theta$ of a BNN having pdf/pmf $p$, and by $P_{x,\theta}$ a generic likelihood on the space $\mathcal{Y}$ of outputs having pdf/pmf $p_{x,\theta}$. The act of computing the posterior from prior $P$ and likelihood $P_{x,\theta}$ using a BNN is designated by $\mathfrak{post}(P, P_{x,\theta})$. The IBNN procedure follows.

---

**Procedure** Imprecise Bayesian Neural Network.

**S1** Specify a *finite* set $\mathcal{P}$ of plausible prior probabilities on the parameters $\theta$ of the neural network, and a *finite* set $\mathcal{L}_{x,\theta}$ of plausible likelihoods.

**S2** Compute posterior $P(\cdot \mid D) \equiv P_D = \mathfrak{post}(P, P_{x,\theta})$ on the parameters $\theta$ of the neural network, for all $P \in \mathcal{P}$ and all $P_{x,\theta} \in \mathcal{L}_{x,\theta}$.

---

Step **S2** performs an element-wise application of Bayes' rule for all the elements of $\mathcal{P}$ and $\mathcal{L}_{x,\theta}$. We obtain a finite set $\mathcal{P}_D$ of posteriors whose cardinality $\#\mathcal{P}_D$ is given by $\#\mathcal{P} \times \#\mathcal{L}_{x,\theta}$. Its convex hull $\mathrm{Conv}\mathcal{P}_D$ is the credal posterior set. By Lemma 2 we have that the upper and lower probabilities of $\mathcal{P}_D$ and $\mathrm{Conv}\mathcal{P}_D$ coincide. In applications, every element of $\mathcal{P}$ should not be obtainable as a convex combination of other elements. This to avoid carrying out redundant operations in Step **S2**. The same holds for the elements of $\mathcal{L}_{x,\theta}$. Furthermore, choosing 2 to 5 priors and likelihoods is usually enough to safely hedge against prior and likelihood misspecification.

Our procedure prescribes the scholar to specify a finite set $\mathcal{P}$ of priors on the parameters of the neural network, and to specify a finite set $\mathcal{L}_{x,\theta}$ of likelihoods (that can be used to capture different architectures of the neural network) that approximate the true data generating process. For every pair of prior $P$ and likelihood $P_{x,\theta}$, the associated posterior $P(\cdot \mid D) \equiv P_D$ on the parameters $\theta$ of the network captures the distribution of these latter, revised according to the available information encapsulated in the likelihood. Eventually, as explained in section 4, we use the posteriors to derive the predictive distributions $\hat{P}$ on the output space $\mathcal{Y}$, which are the distributions of interests for the downstream tasks.

Notice that in the case that $\mathcal{P}$ and $\mathcal{L}_{x,\theta}$ are both not singletons, for all $A \subset \Theta$ the interval $[\underline{P}_D(A), \overline{P}_D(A)]$ is wider than the case when one or other is a singleton. In the limiting case where both are singletons, we retrieve the usual Bayesian updating, so the interval shrinks down to a point.

**Robustness Properties of IBNNs.** We follow the Procedure above to compute the posterior credal set for the parameters of our IBNN. Being trained using credal sets makes IBNNs more robust to distribution shifts than BNNs. To see this, we present the following general result, and then we apply it to our case.

Let $\mathscr{P}$ be a generic set of probabilities, and consider a probability measure $P'$ such that $P' \notin \mathscr{P}$.

**Proposition 4.** *Call $d$ any metric and div any divergence on the space of probability measures of interest. Let $d(\mathscr{P}, P') := \inf_{P \in \mathscr{P}} d(P, P')$ and $div(\mathscr{P} \| P') := \inf_{P \in \mathscr{P}} div(P \| P')$. Then, for all $P \in \mathscr{P}$, $d(\mathscr{P}, P') \leq d(P, P')$ and $div(\mathscr{P} \| P') \leq div(P \| P')$.*

Proposition 4 holds if $\mathscr{P}$ is any set of probabilities, not just a credal sets. In Appendix J, we show that the above result still holds if the elements of $\mathscr{P}$ and $P'$ are defined on Euclidean spaces having different dimensions (Cai & Lim, 2022).

Let us now apply Proposition 4 to IBNNs. Suppose that, when designing a regular BNN, an agent chooses likelihood $\mathbf{P}_{x,\theta}$, while when designing a more general IBNN, they start by specifying a finite set of likelihoods $\mathcal{L}_{x,\theta} = \{P^1_{x,\theta}, \ldots, P^k_{x,\theta} : k \in \mathbb{N}, \theta \in \Theta, x \in \mathcal{X}\}$, and then let the induced credal set $\mathrm{Conv}\mathcal{L}_{x,\theta}$ represent their uncertainty around the sampling model. Assume that $\mathbf{P}_{x,\theta} \in \mathcal{L}_{x,\theta}$ (this means that when designing the regular BNN, the agent chooses arbitrarily which of the elements of $\mathcal{L}_{x,\theta}$ to use) and that the "oracle" data generating process $P^o_{x,\theta}$ is different from $\mathbf{P}_{x,\theta}$, $P^o_{x,\theta} \neq \mathbf{P}_{x,\theta}$, so that we are actually in the presence of distribution shift. Then, we have two cases. (1) If the true sampling model $P^o_{x,\theta}$ belongs to $\mathrm{Conv}\mathcal{L}_{x,\theta}$, then the distance – measured via a metric or a divergence – between $\mathrm{Conv}\mathcal{L}_{x,\theta}$ and $P^o_{x,\theta}$ is 0 while that between $\mathbf{P}_{x,\theta}$ and $P^o_{x,\theta}$ is positive. (2) If $P^o_{x,\theta} \notin \mathrm{Conv}\mathcal{L}_{x,\theta}$, then the distance between $\mathrm{Conv}\mathcal{L}_{x,\theta}$ and $P^o_{x,\theta}$ is no larger than the distance between $\mathbf{P}_{x,\theta}$ and $P^o_{x,\theta}$, no matter (i) which metric or distance we use (Proposition 4), (ii) whether or not $P^o_{x,\theta}$ and the elements of $\mathrm{Conv}\mathcal{L}_{x,\theta}$ are defined on the same Euclidean space (Appendix J, Lemma 16). A visual representation is given in Figure 3.

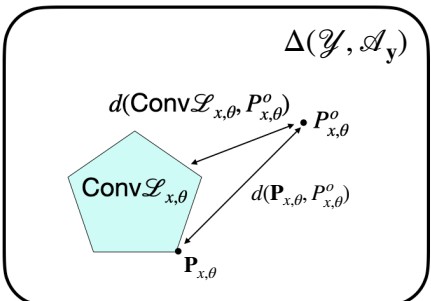

Figure 3: IBNNs are more robust to distribution shifts than regular BNNs. Here $\mathrm{Conv}\mathcal{L}_{x,\theta}$ is the convex hull of five plausible likelihoods, and $d$ denotes a generic metric on the space $\Delta(\mathcal{Y}, \mathcal{A}_{\mathbf{y}})$ of probabilities on $\mathcal{Y}$. We see how $d(\mathrm{Conv}\mathcal{L}_{x,\theta}, P^o_{x,\theta}) < d(\mathbf{P}_{x,\theta}, P^o_{x,\theta})$; if we replace metric $d$ by a generic divergence $div$, the inequality would still hold.

**Computational aspects of IBNNs.** A computational bottleneck of the IBNN procedure appears to be step **S2**, which is a combinatorial task. We have to calculate $\#\mathcal{P} \times \#\mathcal{L}_{x,\theta}$ many posteriors, but this procedure allows us to forego any additional assumptions on the nature of the lower and upper probabilities that are often times required by other imprecise-probabilities-based techniques.[3] Clearly, the procedure is simplified if either $\mathcal{P}$ or $\mathcal{L}_{x,\theta}$ are singletons.

The posteriors in $\mathcal{P}_D$ are typically very high-dimensional and highly non-convex; we use variational inference (VI), outlined in Jospin et al. (2022, Section V), to approximate them in an efficient fashion. That is, we project every $P_D \in \mathcal{P}_D$ onto a set $\mathbb{S}$ of "well-behaved" distributions (e.g. Normals) using the KL divergence. In formulas, $\breve{P}_D = \arg\min_{Q \in \mathbb{S}} KL(Q \| P_D)$. By "well-behaved", we mean that they have to satisfy the conditions in Zhang & Gao (2020, Sections 2 and 3).[4] This ensures that as the sample size goes to infinity, the approximated posteriors converge to the true data generating process. We also point out how despite using a VI approximation for the exact posteriors, by Proposition 4 the credal set of approximated posteriors is closer to the "oracle" posterior $P^o_D$ than any of its elements. As a consequence, working with credal sets leads to VI posterior approximations that are better than the ones resulting from a single BNN, or an ensemble of BNNs, where several BNNs are combined into one.

---

[3]If we are willing to make such assumptions, Theorem 9 in Appendix D shows how to compute the upper posterior using only upper prior and upper likelihood.

[4]We assume that the conditions on the priors and the likelihoods given in Zhang & Gao (2020) are satisfied.

**Remark 5.** *Although highly unlikely in practice, it is theoretically possible that the VI approximation of the (finite) set $\mathcal{P}_D$ of posteriors is a singleton, see Figure 4. While the conditions in Zhang & Gao (2020) guarantee that asymptotically the approximated posteriors coincide with the true data generating process, this typically does not happen with finite-dimensional datasets. As a consequence, obtaining a singleton when projecting $\mathcal{P}_D$ onto $\mathbb{S}$ may result in an underestimation of the uncertainties faced by the scholar. In that case, we either consider a different set – whose elements still satisfy the conditions in Zhang & Gao (2020, Sections 2 and 3) – on which to project $\mathcal{P}_D$ according to the KL divergence, or we use a different "projection operator", that is, a divergence different from the KL. For example, Zhang & Gao (2020) suggest Rényi and $\chi^2$ divergences, or Hellinger and total variation metrics. Alternatively, we can consider a different approximation strategy altogether, for instance the Laplace approximation (Ritter et al., 2018).*

**Remark 6.** *Assume that the "oracle" prior $P^o$ is in the prior credal set $Conv\mathcal{P}$ and that the "oracle" likelihood $P^o_{x,\theta}$ is in the likelihood credal set $Conv\mathcal{L}_{x,\theta}$. Then, it is immediate to see that the "oracle" posterior $P^o_D$ belongs to the posterior credal set $Conv\mathcal{P}_D$. Naturally, this does not imply that the posterior credal set collapses to $P^o_D$. In general, it is unlikely that a finite amount of data is able to completely annihilate all the epistemic uncertainty faced by the agent. What may happen is that if the training set is large enough, $Conv\mathcal{P}_D$ may be inscribed in a ball of small radius around $P^o_D$. This does not mean that we suffer from under-confidence due to larger-than-necessary epistemic uncertainty. Rather, the relative epistemic uncertainty (measured by the difference between prior and posterior uncertainty, divided by the prior uncertainty) drops significantly. In addition, working with sets of prior and likelihoods allows us to hedge against prior and likelihood misspecification: the statements $P^o \in Conv\mathcal{P}$ and $P^o_{x,\theta} \in Conv\mathcal{L}_{x,\theta}$ are either strong assumptions, or very hard to verify.*

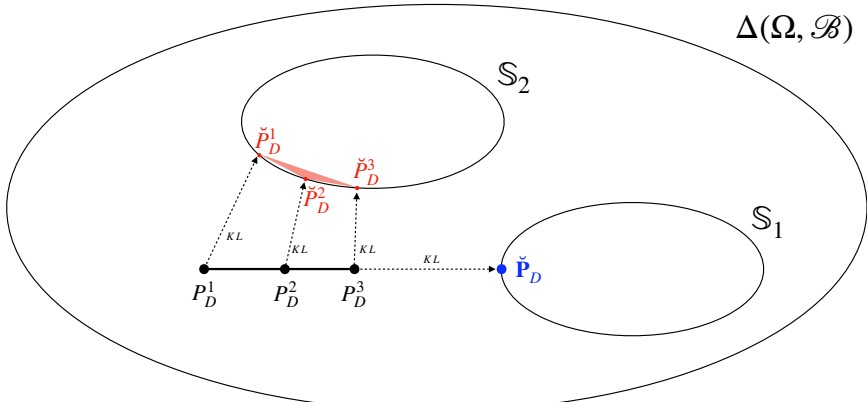

Figure 4: Let $\Delta(\Omega, \mathcal{B})$ denote the space of probability measures on $\Theta$. Let $\mathcal{P}_D = \{P_D^1, P_D^2, P_D^3\}$, so that $Conv\mathcal{P}_D$ is the black segment. Then, if we project the elements of $\mathcal{P}_D$ onto $\mathbb{S}_1$ via the KL divergence, we obtain the same distribution $\check{\mathbf{P}}_D$. This is detrimental to the analysis because such an approximation underestimates epistemic (and possibly also aleatoric) uncertainty faced by the agent. Then, the scholar should specify a different set $\mathbb{S}_2$ of "well-behaved" distributions onto which project the elements of $\mathcal{P}_D$. In the figure, we see that $P_D^1$, $P_D^2$, and $P_D^3$ are projected onto $\mathbb{S}_2$ via the KL divergence to obtain $\check{P}_D^1$, $\check{P}_D^2$, and $\check{P}_D^3$. The convex hull of these latter, captured by the red shaded triangle, represents the variational approximation of $Conv\mathcal{P}_D$.

### 3.2 Aleatoric and epistemic uncertainties

In Hüllermeier & Waegeman (2021), the authors study uncertainty in the context of supervised learning. In particular, they extensively review the existing approaches to quantify aleatoric and epistemic uncertainty (AU and EU, respectively). The former refers to irreducible uncertainty, the variability in the outcome of an experiment which is due to inherently random effects. An example of AU is coin flipping: the data generating process in this experiment has a stochastic component that cannot be reduced by any additional source of information. EU, instead, corresponds to reducible uncertainty, caused by a lack of knowledge about the best model. Continuing the coin example, this corresponds to holding a belief that the coin is biased. After a few tosses, we realize whether or not the coin is in fact biased, or if instead it is fair. Notice how, in the

precise case – that is, when the agent specifies a single distribution – EU is absent. A single probability measure is only able to capture the idea of aleatoric uncertainty, since it represents a case in which the agent knows exactly the true data generating process. This is a well-studied property of credal sets (Hüllermeier & Waegeman, 2021, Page 458). EU should not be confused with the concept of *epistemic probability* (de Finetti, 1974; 1975; Walley, 1991). In the subjective probability literature, epistemic probability can be captured by a single distribution. Its best definition can be found in Walley (1991, Sections 1.3.2 and 2.11.2). There, the author specifies how epistemic probabilities model logical or psychological degrees of partial belief of the agent. We remark, though, how de Finetti and Walley work with finitely additive probabilities, while in this paper we use countably additive probabilities.

Recall that, given a probability measure $P$ on a generic space $\Omega$, the (Shannon) entropy of $P$ is defined as $H(P) := \mathbb{E}[-\log p] = -\int_\Omega \log[p(\omega)]P(\mathrm{d}\omega)$ if $\Omega$ is uncountable, where $p$ denotes the pdf of $P$. If $\Omega$ is at most countable, we have that $H(P) = -\sum_{\omega \in \Omega} P(\{\omega\}) \log[P(\{\omega\})]$. The entropy primarily captures the shape of the distribution, namely its "peakedness" or non-uniformity (Dubois & Hüllermeier, 2007; Hüllermeier & Waegeman, 2021), and hence informs about the predictability of the outcome of a random experiment: the higher its value, the lower the predictability. Now, consider a generic set of probabilities $\mathscr{P}$ on $\Omega$. Then, we can define the imprecise versions of the Shannon entropy as proposed by Abellán et al. (2006); Hüllermeier & Waegeman (2021), $\overline{H}(P) := \sup_{P \in \mathscr{P}} H(P)$ and $\underline{H}(P) := \inf_{P \in \mathscr{P}} H(P)$, called the upper and lower Shannon entropy, respectively.[5] The upper entropy is a measure of total uncertainty since it represents the minimum level of predictability associated with the elements of $\mathscr{P}$. In Abellán et al. (2006); Hüllermeier & Waegeman (2021), the authors posit that it can be decomposed as a sum of aleatoric and epistemic uncertainties, and that this latter can be specified as the difference between upper and lower entropy, thus obtaining

$$\underbrace{\overline{H}(P)}_{\text{total uncertainty}} = \underbrace{\underline{H}(P)}_{\text{aleatoric uncertainty}} + \underbrace{\left[\overline{H}(P) - \underline{H}(P)\right]}_{\text{epistemic uncertainty}}.$$

We have the following proposition.

**Proposition 7.** *Let $\Pi, \Pi'$ be sets of probability measures as the ones considered in Remark 1. Then, $\sup_{P \in \Pi} H(P) = \overline{H}(P) \leq \overline{H}(P') = \sup_{P \in \Pi'} H(P)$.*

Proposition 7 tells us that the upper entropy of the extreme elements in $\Pi = \mathrm{ex}\Pi'$ is a lower bound for the upper entropy of the whole credal set $\Pi'$. An immediate consequence of Proposition 7 is that the lower entropy of the extreme elements in $\Pi$ is an upper bound for the lower entropy of the whole credal set $\Pi'$.

In the context of IBNNs, we are especially interested in EU and AU associated with the *predictive* credal set, introduced in the next section. It is the set of distributions on the output set $\mathcal{Y}$ induced by the posterior distributions in $\mathcal{P}_D$. We also point out a salient feature of this approach: the EU and AU computed from the predictive credal set embed uncertainty comparable to an uncountably infinite ensemble of BNNs, i.e. an ensemble of BNNs of cardinality $\aleph_1$, despite the simple and intuitive mathematics over a finite set. They are not merely pessimistic bounds to the uncertainties associated with a finite ensemble of BNNs.

## 4 Practical aspects

In this section, we first illustrate how to elicit credal prior and likelihood sets that are needed for step **S1** of the procedure in section 3.1. Then, we describe how IBNN are instrumental in specifying a set of outputs that enjoys probabilistic guarantees.

Outlined in Jospin et al. (2022, Sections IV-B and IV-C1), for classification, the standard process for BNNs involves

- A Normal prior with zero mean and diagonal covariance $\sigma^2 I$ on the coefficients of the network, that is, $p(\theta) = \mathcal{N}(0, \sigma^2 I)$. In the context of IBNNs, we could specify e.g. $\mathcal{P} = \{P : p(\theta) = \mathcal{N}(\mu, \sigma^2 I), \mu \in \{\mu_-, \mathbf{0}, \mu_+\}, \sigma^2 \in \{3, 7\}\}$. That is, the extreme elements of the prior credal set are five independent Normals having different levels of "fatness" of the tails, and centered at a vector $\mu_+$ having positive

---

[5]In Appendix E, we provide bounds to the values of upper and lower entropy.

entries, a vector $\mu_-$ having negative entries, and a vector $\mathbf{0}$ having entries equal to 0. They capture the ideas of positive bias, negative bias, and no bias of the coefficients, respectively. This is done to hedge against possible prior misspecification.

- A Categorical likelihood, $p(y \mid x, \theta) = \text{Cat}(\Phi_\theta(x))$, whose parameter is given by the output of a functional model $\Phi_\theta$. In the context of IBNNs, we could specify the set of extreme elements of the likelihood credal set as $\mathcal{L}_{x,\theta} = \{P_{x,\theta} : p_{x,\theta}(y) = \text{Cat}(\Phi_{s,\theta}(x)), s \in \mathcal{S}\}$, where $\mathcal{S} = \{1, \ldots, S\} \subset \mathbb{N}$ is a generic index set. Specifying set $\mathcal{L}_{x,\theta}$, then, corresponds to eliciting a finite number $S$ of possible (parametrized) architectures for the neural network, $\Phi_{s,\theta}$, $s \in \mathcal{S} = \{1, \ldots, S\}$, and obtain, as a consequence, $S$ categorical distributions $\{\text{Cat}(\Phi_{s,\theta}(x))\}_{s \in \mathcal{S}}$. This captures the ambiguity around the true data generating process faced by the agent, and allows them to hedge against likelihood misspecification.

More in general, we can use the priors and likelihoods that better fit the type of analysis we are performing. For example, for the choice of the priors we refer to Fortuin et al. (2021), where the authors study the problem of selecting the right type of prior for BNNs. We compute the $N := \#\mathcal{P} \times \#\mathcal{L}_{x,\theta}$ posteriors $\mathfrak{post}(P, P_{x,\theta})$, for all $P \in \mathcal{P}$ and all $P_{x,\theta} \in \mathcal{L}_{x,\theta}$. They are the elements of set $\mathcal{P}_D$, whose convex hull $\text{Conv}\mathcal{P}_D$ represents the posterior credal set. The posteriors $P_D \in \mathcal{P}_D$ are distributions on the parameter space $\Theta$.[6] Every such posterior induces a distribution on the output space $\mathcal{Y}$ via the predictive distribution

$$p(\tilde{y} \mid \tilde{x}, x_1, y_1, \ldots, x_n, y_n) = \int_\Theta p(\tilde{y} \mid \theta, \tilde{x}) \cdot p(\theta \mid x_1, y_1, \ldots, x_n, y_n) \, \mathrm{d}\theta,$$

where $\tilde{x}$ is a new input and $\tilde{y}$ is the associated output (see Appendix F for more details). In symbols, we can write $P_D \rightsquigarrow \hat{P}$, for all $P_D \in \mathcal{P}_D$, and let $\hat{\mathcal{P}} := \{\hat{P}_1, \ldots, \hat{P}_N\}$. Its convex hull $\text{Conv}\hat{\mathcal{P}}$ is the *predictive credal set*. To save notation, we write $\hat{p}_k(\cdot) \equiv p_k(\cdot \mid \tilde{x}, x_1, y_1, \ldots, x_n, y_n)$, for all $k \in \{1, \ldots, N\}$. Now, for every such distribution we compute the $\alpha$-*level Highest Density Region (HDR)* $R(\hat{p}_k^\alpha)$, $\alpha \in (0, 1)$. As defined in Coolen (1992, Section 1), it is a subset of the output space $\mathcal{Y}$ such that

$$\int_{R(\hat{p}_k^\alpha)} \hat{p}_k(y)\mathrm{d}y \geq 1 - \alpha \quad \text{and} \quad \int_{R(\hat{p}_k^\alpha)} \mathrm{d}y \text{ is a minimum.}$$

We need $\int_{R(\hat{p}_k^\alpha)} \mathrm{d}y$ to be a minimum because we want $R(\hat{p}_k^\alpha)$ to be the smallest possible region that gives us the desired probabilistic coverage. Equivalently, we can write that $R(\hat{p}_k^\alpha) = \{y \in \mathcal{Y} : \hat{p}_k(y) \geq \hat{p}_k^\alpha\}$, where $\hat{p}_k^\alpha$ is a constant value. In particular, it is the largest constant such that $\hat{P}_k[y \in R(\hat{p}_k^\alpha)] \geq 1 - \alpha$ (Hyndman, 1996). In dimension 1, $R(\hat{p}_k^\alpha)$ can be interpreted as the narrowest interval – or union of intervals – in which the value of the (true) output falls with probability of at least $1 - \alpha$ according to distribution $\hat{P}_k$. As we can see, HDR's are a Bayesian counterpart of confidence intervals. We give a simple visual example in Figure 5.

Finally, compute the $\alpha$-*level Imprecise Highest Density Region (IHDR)* $IR_\alpha := \cup_{k=1}^N R(\hat{p}_k^\alpha)$. By taking the union of the HDR's, we ensure that all the probability measures in the predictive credal set $\text{Conv}\hat{\mathcal{P}}$ assign probability of at least $1 - \alpha$ to the event $\{y \in IR_\alpha\}$; this is a consequence of Lemma 2. In turn, this implies that $\underline{P}(y \in IR_\alpha) = \inf_{k \in \{1, \ldots, N\}} \hat{P}_k(y \in IR_\alpha) \geq 1 - \alpha$. This can be interpreted as the event $A = \{\text{The (true) output belongs to } IR_\alpha\}$ having lower probability of at least $1 - \alpha$, a probabilistic guarantee for the set of outputs $IR_\alpha$ generated by our procedure.

This method allows to quantify and control the difference between the upper and lower probabilities of $IR_\alpha$. To see this, notice that $\overline{P}(y \in IR_\alpha) \leq 1$, so $\overline{P}(y \in IR_\alpha) - \underline{P}(y \in IR_\alpha) \leq \alpha$. Notice also that the size of $IR_\alpha$ is an increasing function of both AU and EU. As a consequence, it is related, but it is not equal, to the AU the agent faces. If we want to avoid to perform the procedure only to discover that $IR_\alpha$ is "too big", then we can add an "AU check" at the beginning. This, together with computing $IR_\alpha$ in a classification setting, is explored in Appendices G and H.

We conclude with a remark. We first point out that in applications the predictive density $\hat{p}$ is computed using the VI approximation $\check{p}$ of the posterior density, that is, by solving $\int_\Theta p(\tilde{y} \mid \theta, \tilde{x}) \cdot \check{p}(\theta \mid x_1, y_1, \ldots, x_n, y_n) \, \mathrm{d}\theta$.

---

[6]Assume for simplicity that they all have density with respect to some ($\sigma$-finite) dominating measure.

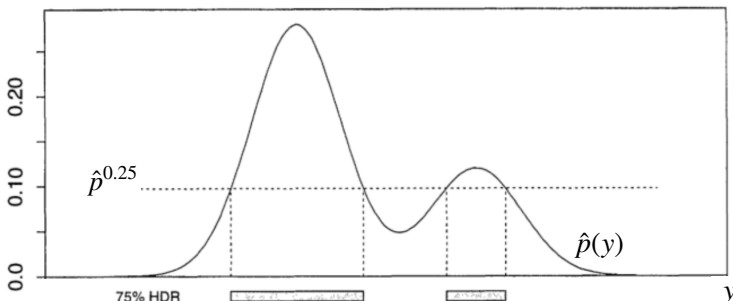

Figure 5: The 0.25-HDR from a Normal Mixture density. This picture is a replica of Hyndman (1996, Figure 1). The geometric representation of "75% probability according to $\hat{P}$" is the area between the pdf curve $\hat{p}(y)$ and the horizontal bar corresponding to $\hat{p}^{0.25}$. A higher probability coverage (according to $\hat{P}$) would correspond to a lower constant, so $\hat{p}^{\alpha} < \hat{p}^{0.25}$, for all $\alpha < 0.25$. In the limit, we recover 100% coverage at $\hat{p}^0 = 0$.

This allows to obtain $\hat{p}$ even when the exact posterior is intractable. Second, we observe that the scholar is mostly interested in the AU and EU associated with the credal set $\text{Conv}\hat{\mathcal{P}}$ of predictive distributions. This because they are ultimately interested in reporting the uncertainty around the predicted outputs given a new input in the problem at hand, more than the uncertainty on the parameters of the BNNs and on the likelihoods. Finally, a tacit assumption is that the EU and AU associated with the predictive credal set $\text{Conv}\hat{\mathcal{P}}$ are similar whether we compute its elements using the exact posteriors or their VI approximation. This is needed because in real-world applications the agent would not be able to make the ambiguity around the correct prior and likelihood percolate towards downstream tasks without using variational inference (or other types of approximations).

## 5 Experiments

In this section, we focus on the downstream task performance of IBNNs. We show that they perform better than an ensemble of BNNs (EBNN) – introduced formally in the next paragraph. To demonstrate the utility of the proposed method, we study the behavior of certain safety-critical settings under distribution shifts and its ramifications. One, for motion prediction in autonomous driving scenarios, and two, to model blood glucose and insulin dynamics for artificial pancreas control.

In Egele et al. (2021), the authors pursue the same heuristic endeavor as we do, but take an ensemble route. They too consider different BNNs, but instead of keeping them separate and use them to build a predictive credal set, they average them out. Similar to theirs, we elicit the following procedure, that we call ensemble of BNNs (EBNN). Consider $k$ different BNNs, and compute the posterior distribution on the parameters. They induce $k$ predictive distributions on the output space $\mathcal{Y}$, each having mean $\mu_j$ and variance $\sigma_j^2$. We call *EBNN distribution* $P_{\text{ens}}$ a Normal having mean $\mu_{\text{ens}} = 1/k \sum_{j=1}^k \mu_j$ and covariance matrix $\sigma_{\text{ens}}^2 I$, where $\sigma_{\text{ens}}^2 = 1/k \sum_{j=1}^k \sigma_j^2 + 1/(k-1) \sum_{j=1}^k (\mu_j - \mu_{\text{ens}})^2$. We use the $\alpha$-level HDR associated with $P_{\text{ens}}$ as a benchmark for our $IR_\alpha$. Being rooted in IP theory, IBNNs are a better justified procedure than EBNN from a theoretical standpoint. In this section, we show with two applications that their implementation performs better than EBNN. We do not consider Bayesian Model Averaging (BMA) as a baseline for IBNNs because BMA suffers from pitfalls when used in the context of Bayesian neural networks (Izmailov et al., 2021a). In addition, we do not compare IBNNs against belief tracking techniques because these latter require extra assumptions that IBNNs do not, see Appendix L.

**Uncertainty Quantification Benchmark Experiments.** In Appendix I, we include details of experiments where we train IBNNs for image classification tasks for standard datasets like CIFAR10 (Krizhevsky et al., 2009), SVHN (Netzer et al., 2011), Fashion-MNIST (Xiao et al., 2017), and MNIST (Lecun et al., 1998). There, we discuss how IBNNs are better than EBNN at disentangling AU and EU, and at quantifying them. While for IBNNs an increase in the corruption of the dataset corresponds to an increase in both EU

and AU, for EBNN the AU either stays constant, or it decreases. This counterintuitive – and erroneous – behavior makes us conclude that we are able to capture the uncertainties better than the baseline.

## 5.1 Motion Prediction for Autonomous Racing

In this case study, we demonstrate the utility of IBNNs for motion prediction in autonomous driving scenarios. An important challenge in autonomous driving is understanding the intent of other agents and predicting their future trajectories to allow for safety-aware planning. In autonomous racing, where control is pushed to the dynamical limits, accurate and robust predictions are even more essential for outperforming opponent agents while assuring safety. Again, IBNNs provide a straightforward method for quantifying uncertainty and deriving robust prediction regions for anticipating an agent's behavior.

We use the problem settings in Tumu et al. (2023) to define the problem of obtaining prediction sets for future positions of an autonomous racing agent. Our results show that the prediction regions have improved coverage when compared to EBNN. These results hold in both in-distribution and out-of-distribution settings, which are described below.

**Problem.** Let $O^i(t, l) \equiv O^i = \{\pi_{t-l}^i, \ldots, \pi_t^i\}$ denote the $i$-th trajectory instance of an agent at time $t$, consisting of the observed positions from time $t - l$ up to time $t$. Let then $C^i$ be a time-invariant context variable. Let also $F^i(t, h) \equiv F^i = \{\pi_{t+1}^i, \ldots, \pi_{t+h}^i\}$ be the collection of the next $h$ future positions. We wish to obtain a model $M$ that predicts region $\mathcal{R}_\alpha$ with probabilistic guarantees. In particular, for EBNN $\mathcal{R}_\alpha$ is the $\alpha$-level HDR of $P_{\text{ens}}$, so that $P_{\text{ens}}(F^i \in \mathcal{R}_\alpha) \geq 1 - \alpha$, while for IBNN $\mathcal{R}_\alpha = IR_\alpha$, so that $\underline{\hat{P}}(F^i \in \mathcal{R}_\alpha) \geq 1 - \alpha$.

The dataset consists of instances of $(O^i, F^i)$ divided into a training set $\mathcal{D}_{\text{train}}$ and a testing set $\mathcal{D}_{\text{test}}$. We train an uncertainty-aware model on $\mathcal{D}_{\text{train}}$ that computes the triplet $(F_l^i, F_m^i, F_u^i) = M(O^i, C^i)$ where $F_l^i$, $F_u^i$, $F_m^i$ are the lower, upper, and mean predictions of the future positions, respectively.

The dataset $\mathcal{D}_{\text{all}}$ is created by collecting simulated trajectories of autonomous race cars in the F1Tenth-Gym (O'Kelly et al., 2020) (details in Tumu et al. (2023)). As shown in Figure 6, different racing lines were utilized including the center, right, left, and optimal racing line for the Spielberg track.

Figure 6: Motion Prediction for F1Tenth-Gym Environment (O'Kelly et al., 2020). Data is collected by simulating various racing lines on the Spielberg Track.

We denote these by $\mathcal{D}_{\text{center}}$, $\mathcal{D}_{\text{right}}$, $\mathcal{D}_{\text{left}}$, and $\mathcal{D}_{\text{race}}$, respectively. Position $\pi$ is a vector $\pi = (a, b, \vartheta, v)^\top$, where $a$ and $b$ are coordinates in a 2-dimensional Euclidean space, and $\vartheta$ and $v$ are the heading and speed, respectively. In total, the $\mathcal{D}_{\text{all}}$ consists of 34686 train instances, 4336 validation instances, and 4336 test instances.

**In-distribution vs. Out-of-distribution.** We consider the prediction task to be in-distribution when $\mathcal{D}_{\text{train}}, \mathcal{D}_{\text{test}} \subset \mathcal{D}_{\text{all}}$. It is out-of-distribution (OOD) when $\mathcal{D}_{\text{train}} \subset \mathcal{D}_{\text{center}} \cup \mathcal{D}_{\text{right}} \cup \mathcal{D}_{\text{left}}$ and $\mathcal{D}_{\text{test}} \subset \mathcal{D}_{\text{race}}$.

**Metrics.** We train the ensemble of BNNs and the IBNN models, $M_{\text{ens}}$ and $M_{\text{IBNN}}$ respectively, using the same architecture and different seeds. We compare the performance with respect to the test set by

computing the single-step coverage, where each prediction time-step is treated independently, and the multi-step coverage, which considers the entire $h$-step prediction.

Figure 7 (a) depicts a sample of the in-distribution evaluation for each of the models. For a given trajectory, the red boxes indicate when the prediction region did not cover the actual trajectory at that time-step. Qualitatively, $M_{\text{IBNN}}$ has less missed timesteps when compared to $M_{\text{ens}}$. Table 1 shows that IBNNs perform better in terms of both one-step and multi-step coverage. Similar results can be observed for the OOD scenario. There, all models were trained on racing lines which are predominantly parallel to the track curvature. As a consequence, when the test set consists of instances with higher curvatures, the overall coverage of all models degrades. This can be seen in Figure 7 (b), where the prediction of the models (orange) tends to be straight while the actual trajectory is more curved (green). Despite this, the figure and the coverage metrics in Table 1 show how IBNN exhibits a more robust behavior.

| | In-Distribution Results | | | | | |
| | Ensemble | | | IBNN | | |
| $1 - \alpha$ | 0.9 | 0.95 | 0.99 | 0.9 | 0.95 | 0.99 |
|---|---|---|---|---|---|---|
| One-step | 0.962 | 0.980 | 0.992 | **0.992** | **0.995** | **0.997** |
| Multi-step | 0.638 | 0.826 | 0.937 | **0.914** | **0.948** | **0.979** |
| | Out-of-Distribution Results | | | | | |
| | Ensemble | | | IBNN | | |
| $1 - \alpha$ | 0.9 | 0.95 | 0.99 | 0.9 | 0.95 | 0.99 |
| One-step | 0.919 | 0.950 | 0.980 | **0.979** | **0.988** | **0.995** |
| Multi-step | 0.532 | 0.703 | 0.860 | **0.825** | **0.884** | **0.943** |

Table 1: F1Tenth coverage results. We report one-step coverage and multi-step coverage across 3 different values of $\alpha$. IBNNs exceed coverage of EBNNs in all settings.

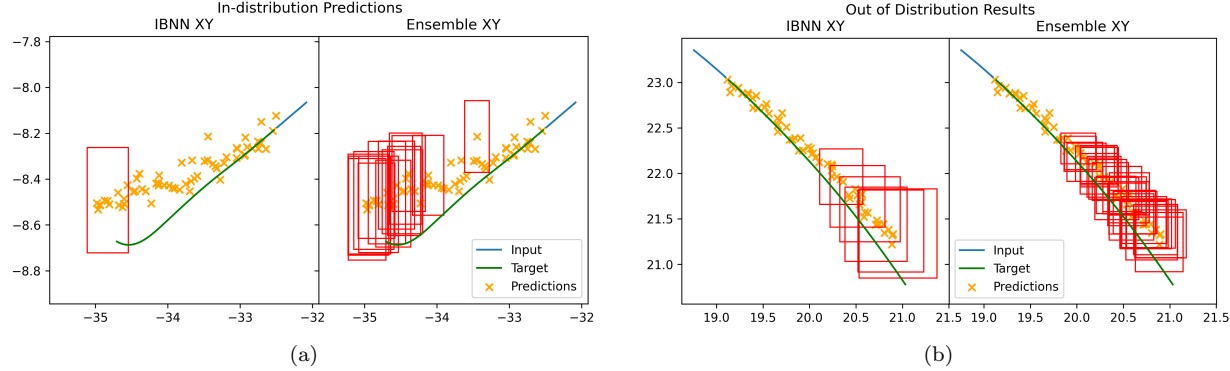

(a)  (b)

Figure 7: **Left:** F1Tenth in-distribution Results. Given an input of past observations, IBNNs exhibit better coverage of the future target trajectory. Predictions which do not cover the target within the desired $1 - \alpha$ level are indicated in red. **Right:** F1Tenth Out-of-distribution (OOD) Results. Robust performance is exhibited by IBNNs when compared to EBNN in OOD settings.

## 5.2 Artificial Pancreas Control

**Overall Setup.** In this next case study we consider the problem of data-driven control of human blood glucose-insulin dynamics, using an artificial pancreas system, Figure 8.

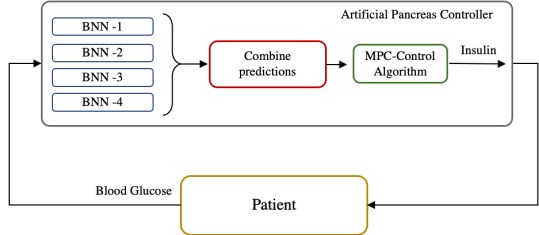

Figure 8: The Bayesian neural networks predict a future blood glucose value. These individual predictions are combined to get a robust estimate of the true value as an interval. This is used by the MPC control algorithm to recommend insulin dosage for the patient. The patient block in our experiment is simulated using the virtual patient models from the UVa-Padova simulator.

External insulin delivery is accomplished by using an insulin pump controlled by the artificial pancreas software, which attempts to regulate the blood-glucose (BG) level of the patient within the euglycemic range of $[70, 180]mg/dl$ (Kushner et al., 2018). Levels below $70mg/dl$ lead to hypoglycemia, which can lead to loss of consciousness, coma or even death. On the other hand, levels above $300mg/dl$ lead to a condition called the ketoacidosis, where the body can break down fat due to lack of insulin, and lead to build up of ketones. In order to treat this situation, patients receive external insulin delivery through insulin pumps. Artificial Pancreas (AP) systems can remedy this situation by measuring the blood glucose level, and automatically injecting insulin into the blood stream. Thus, we define the *unsafe regions* of the space as $G(t) \in (-\infty, 70) \cup (300, \infty)$, where $G(t)$ is the BG value at time $t$. This is the shaded region in Figure 9.

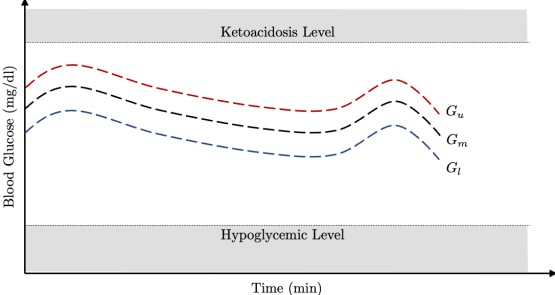

Figure 9: Starting from an initial glucose value, the task of the artificial pancreas controller is to maintain the blood glucose value within safe operating limits using insulin as a mode of control.

**Neural Network Models and Controller.** Deep neural networks are effective in capturing the BG-insulin dynamics for personalized medical devices (Kushner et al., 2018). This allows for improved device performance. Even though standard feedforward neural networks can be used, Bayesian neural networks (BNN), and especially a collection of multiple BNNs, offer a better alternative towards uncertainty aware predictions. Here, we test the ramifications of these prediction sets, when used inside an online receding horizon control scheme for insulin delivery. We use the standard MPC control scheme for this purpose, well-known in the literature (Dutta et al., 2018). More formally, let $G(t)$ and $I(t) \equiv I_t$ be the blood-glucose and insulin values at time $t$, respectively. We denote the finite length trajectory of length $H$ as $\overleftarrow{G}_H(t) := [G(t - H + 1), \dots, G(t)]$, and $\overleftarrow{I}_H(t) := [I(t - H + 1), \dots, I(t)]$. An uncertainty aware model $M$ computes the triplet $(G_l(t+l), G_m(t+l), G_u(t+l)) = M(\overleftarrow{G}_H(t), \overleftarrow{I}_H(t))$, where $G_m$ is the mean prediction output, and $G_l, G_u$ are the lower and upper predictions of the glucose value, respectively. By design, it is true that $G_l \leq G_m \leq G_u$. A model predictive control algorithm – whose cost function we denote by $J$ – solves $\arg\min_{I_0, I_1, \dots, I_{k-1}} \sum_{i=0}^{k-1} J(M(\overleftarrow{G}_H(t+i), \overleftarrow{I}_H(t+i)))$.

After every time step, the control algorithm picks the first insulin input $I_0$ as the insulin bolus for the patient, and discards the rest. Cost function $J$ takes into account three factors, (i) Distance of the mean prediction

level $G_m$ at each time step from a target value of $120mg/dl$, (ii) Distance of upper and lower predictions ($G_u$ and $G_l$) from the unsafe regions of the state space $G(t) > 300$ and $G(t) < 70$, and (iii) Total insulin injected $\sum_{t=0}^{k-1} I_t$.

Starting with some initial glucose value $G(0)$, we measure the performance of the artificial pancreas controller as the fraction of time it spends in the unsafe regions,

$$ t_{\text{unsafe}} = \frac{1}{L} \sum_{t=1}^{L} \mathbb{1} \left\{ G(t) \in (-\infty, 70) \cup (300, \infty) \right\}, $$

where $\mathbb{1}\{\cdot\}$ denotes the indicator function. A lower value is more desirable. We compare EBNN and IBNNs as different realizations of the model $M$.

**Distribution Shift using Meals.** A well known problem with learnt models is distribution shift. Bayesian neural networks can address this issue by apprising the end user of the increased uncertainty. For regression models of the type described above, this appears as larger prediction intervals $[G_l, G_u]$. The artificial pancreas controller can run into this situation in the following way: the insulin-glucose time series data collected for training the data-driven model $M$ can be without meals, while at test time the patient can have meals. This creates a distribution shift between the training and test time data. Fortunately, the UVa-Padova simulator (Dalla Man et al., 2013) allows us to create datasets with and without meal inputs. In this case study, the training data was obtained by randomly initializing the BG value in the range $[120, 190]$, and simulating the patient for 720 minutes. The controller was executed at 5 minutes intervals. At test time the patient was supplied meals at specific time intervals (for details, see Appendix K). This creates a significant distribution shift since meals are effectively an unknown variable which can affect the system state. However, from the controller's perspective this is practical, since patients can have unannounced meals.

**Results and Discussion.** To capture the difference in performance between EBNN and IBNN, we compute $\text{Perf}_{\text{diff}} := (t_{\text{unsafe}}^{\text{EBNN}} - t_{\text{unsafe}}^{\text{IBNN}})/t_{\text{unsafe}}^{\text{EBNN}}$. Both $t_{\text{unsafe}}^{\text{EBNN}}$ and $t_{\text{unsafe}}^{\text{IBNN}}$ depend on interval $[G_l, G_u]$; for EBNN, this latter corresponds to the $\alpha$-level HDR associated with EBNN distribution $P_{\text{ens}}$, while for IBNN it corresponds to the IHDR $IR_\alpha$. We consider one case in which an IBNN is trained using a credal prior set and only one likelihood (we choose different seeds which initialize the prior distributions but we keep the same architecture for the BNNs), and another case in which we do the opposite (we use the same seed and different architectures).

We report $\text{Perf}_{\text{diff}}$, across different choices in Table 2. We observe that a more conservative estimate, as informed by the IBNN model as compared to the EBNN framework, results in controllers which respect the safety limits better. To see that IBNNs are more conservative than EBNNs, notice that when combining predictive distributions from multiple BNNs, an IBNN combines the predictions via a finitely generated credal set (FGCS) whose extrema are the individual distributions. On the contrary, EBNNs take an average of the individual distributions to compute the ensemble distribution $P_{\text{ens}}$. The union of the HDRs of the predictive distributions is more conservative (i.e., broader) than the HDR of the single ensemble distribution. While the approach by EBNNs seems like a reasonable choice on the surface, it falls short in capturing the uncertainty necessary for the downstream task. For more details on this case study, see Appendix K.

| $1 - \alpha$ | 0.9 | 0.95 | 0.99 |
|---|---|---|---|
| Varying Seeds | **2.3**% | **3.5**% | **5.2**% |
| Varying Architectures | **0.5**% | -3.8% | **4.4**% |

Table 2: We report the performance improvements when using IBNNs as compared to EBNNs across 3 different values of $\alpha$. Row 1 corresponds to the case where the individual BNNs are trained with different seeds for the prior distribution; and Row 2 is the case when the BNNs have different architectures.

## 6 Related Work

In Corani et al. (2012), the authors introduce credal classifiers (CCs) as a generalization of classifiers based on Bayesian networks. Unlike CCs, IBNNs do not require independence assumptions between non-descendant, non-parent variables. In addition, IBNNs avoid NP-hard complexity issues of searching for optimal structure in the space of Bayesian networks (Chickering et al., 2004). In Manchingal & Cuzzolin (2022), an epistemic convolutional neural network (ECNN) is developed that explicitly models the epistemic uncertainty induced by training data of limited size and quality. A clear distinction is that ECNNs measure uncertainty in target-level representations whereas IBNNs identify the uncertainty measure on the output space. Despite the merit of their work, we believe IBNNs achieve greater generality, since they are able to quantify aleatoric and epistemic uncertainty and are applicable to problems beyond classification. For a review of the state of the art concerning the distinction between EU and AU we refer to Hüllermeier & Waegeman (2021) and to Manchingal & Cuzzolin (2022). We also point out how IBNNs have been recently used to solve prior-likelihood conflicts in Bayesian statistics (Marquardt et al., 2023). Further references can be found in Appendix M.

## 7 Conclusion

We presented IBNNs, a generalization of BNNs that allows to distinguish between AU and EU, and to quantify them. We showed how they can be used to specify a set of outputs that enjoys probabilistic guarantees, and we applied them to safety-critical settings.

We point out how a region that improves $IR_\alpha$, meaning that it would be tighter, is the following: $IR'_\alpha = \{y \in \mathcal{Y} : \underline{\hat{p}}(y) \geq \underline{\hat{p}}^\alpha\}$, where $\underline{\hat{p}}$ is the lower pdf/pmf evaluated at $y$, and $\underline{\hat{p}}^\alpha$ is the largest constant such that $\underline{\hat{P}}(y \in IR'_\alpha) \geq 1 - \alpha$. The problem with $IR'_\alpha$ is that, while the highest density regions associated with the predictive distributions can be computed using off-the-shelf tools, calculating $\underline{\hat{p}}$ and $\underline{\hat{p}}^\alpha$ would have been much more computationally expensive. In addition, it would have required to come up with a new technique to find $\underline{\hat{p}}$ and $\underline{\hat{p}}^\alpha$. We defer studying this to future work.

We also plan to apply IBNNs to continual learning (CL) to overcome the curse of dimensionality and to capture an agent's preference over the tasks to perform.

Furthermore, we intend to relate IBNNs to Bayesian model selection (BMS) (Ghosh et al., 2019). This latter suffers from the same problem as "regular" Bayesian inference. That is, while it tries to come up with a sophisticate prior that induces shrinkage, it still relies on the "correctness" of that prior, i.e. on correctly specifying the prior's parameters. In the future, an interesting way of combining IBNNs with BMS will be to use a finite number of regularized horseshoe priors, as suggested by Ghosh et al. (2019, Section 3.2), as extreme elements of the prior credal set.

We also call attention to the fact that an IBNN is a model-based approach. The relationship with model-free approaches such as conformal prediction will be the object of future studies. In particular, we are interested in finding in which cases IHDRs are narrower than conformal regions, and vice versa.

Finally, we point out how one possible way of easing the burden of the combinatorial task in step **S2** of the IBNN procedure is to specify a prior credal set whose size strikes the perfect balance between being "vague enough" so that we do not underestimate the EU, and being "small enough" so that the IBNN is actually implementable. We suspect conjugacy of the priors may play a key role in this endeavor. Because of its centrality, we defer the study of "optimal prior credal sets" to future work.

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
