# OpenReview forum: "Imprecise Bayesian Neural Networks"
_TMLR — Rejected by TMLR_

### Review · Reviewer_tXse · 2023-11-09

**Summary Of Contributions:**

This paper presents an alternative ensembling strategy for Bayesian Neural Networks (BNN), leveraging theoretical foundations from imprecise probability. This enables a more cohesive uncertainty quantification approach as the ensemble is not averaged but is rather used as a set from which combinations/intersections of the posterior estimates of the model parameters can be made to improve the downstream task.

With this, the claims that I was able to extract from the paper was that the proposed Imperfect BNN (IBNN) provides:
 - a clear approach (and theoretical justification) for estimating both aleatoric and epistemic uncertainties
 - that these uncertainties provide better coverage and calibration of uncertain predictive qualities
 - This improved uncertainty quantification leads to improved downstream performance.

**Audience:**

No

**Broader Impact Concerns:**

None as far as I understand the intended contributions of the paper.

**Claims And Evidence:**

No

**Requested Changes:**

There needs to be more care taken to orient the reader to what the actual focus of the paper is. The experiments should align with this focus to validate the hypotheses and claims made at the outset of the paper. As currently written, the paper assumes that the reader understands or knows what the implications of aleatoric and epistemic uncertainty are, what the limitations of BNNs are, etc. There needs to be a thorough revision of the paper's construction and the level of detail needed to properly provide the foundation for the work.

Also, it seems that the authors have only focused on the BNN literature for uncertainty quantification. There were vague references to conformal learning or calibrating the uncertainty of NNs but there has been extensive work looking at distribution free or evidence-based uncertainty quantification. Specifically I refer the authors to the literature on Prior Networks (Sensoy, et al 2018), Posterior Networks (Charpentier et al, 2020; 2021; 2022), Deep Evidential Regression (Amini, et al 2020) among others.  Among these papers, there is ample opportunity to have other benchmarks that do uncertainty quantification (and resolve or disentangle aleatoric and epistemic uncertainty).

If the major claims of the paper are that by using IP to better ensemble BNNs, then it should be written that way. I found that the experiments presented a bit of a surprise in that the same 4 BNNs were used for EBNN and IBNN but that the specific way that they were combined was the major contribution. This was not the expectation based on the writing and language put forward at the beginning of the paper.

Additionally, from the outset of the paper it seemed that the focus would be on disentangling and providing better estimates of the aleatoric and epistemic uncertainties. This was reinforced with the complete (and really good!) analysis using classification models on benchmark datasets. As a quick aside, the analysis included in Section I absolutely should be in the main body of the paper. But this level of analysis was not continued through the rest of the paper to it's detriment. I would highly recommend that further experimental analyses are included to isolate why IBNNs are preferred for the path prediction task as well as the glucose-insulin dynamics task. This paper is on the precipice of being really interesting and complete. The poor structure and incomplete experiments are major factors of what is keeping it from being accepted for publication.

**Strengths And Weaknesses:**

`Strengths`

The paper is well justified in it's development of the credal sets and the underlying motivation for the proposed IBNN as an alternative to more traditional ensembles of BNNs. I felt that the background section (Section 2) was well composed and helped orient the reader to understand the necessary principles to follow the development of IBNNs in Section 3.

Great care was taken in the presentation of figures to help illustrate the finer technical details contained in the paper. I felt that these were especially helpful for me as I reviewed the work.

The experiments used in the paper are appropriately scoped and are set-up to properly test the utility of the proposed IBNN (Note very specific concerns about the completeness of the experimental analyses below).

`Weaknesses`

The setting and practical usage of the proposed IBNN is not made clear. It comes across as though the authors are trying to hide that the IBNN is merely a non-condensed ensemble of BNN models. The writing is elevated at the beginning to try and differentiate IBNNs from EBNNs to the point that it is attempting to present IBNNs as a wholly separate concept. This doesn't strengthen the perception of the contributions/claims made in this paper and only serves to confuse the reader. I believe that a more direct and honest discussion of what IBNNs are would greatly improve the readability and reception of the work. Additionally, there is fairly little description of how the IBNN procedure is actually implemented and how predictions are actually made. There is a lot of detail obfuscated by the reliance on the mathematical equations to communicate what is actually being done (Remark 5 and 6 add further confusion).

On this note, it is unclear what the proposed IBNN is being presented for. Is it only improved uncertainty quantification? Do we not care about model performance? The accuracy and efficacy of the models seems to be an afterthought.

Section 4 is not very clearly articulated and could stand to be more direct about how the HDR and IHDR's are computed. A major clarification is needed to make the experiments more clearly understood.

The assumed focus (based on the first three sections of the paper) on representing aleatoric and epistemic uncertainties disappears after a vague mention of the experiments contained in Section I of the Supplement. This gives an impression that the major claims of the paper have become an afterthought. There is very little consistency in what the expected takeaways from each experiment are afterwards. With this being said, I feel that the strongest experimental analysis that validates the claims made in this paper are the classification results in Section I of the Supplement. It is unfortunate that they were relegated to not be a major part of the main body of the paper.

The additional experiments contained in Section 5 are woefully incomplete and have some shifted emphasis away from the stated advantages and claims of the paper (or at least how I understood them). The secondary or downstream effects of improved uncertainty quantification are great. But there is a major departure from what I felt that the paper was setting up in the prior sections. The analysis of the experiments is only kept at a surface level and there are not any real insights drawn from why and how the IBNN approach improves over EBNN. It is perhaps that I didn't fully appreciate how the IHDR is composed to differentiate from HDR. However, the lack of thorough and rigorous experimental results for the problems presented in the main paper are concerning. I was especially disappointed by the lack of any real experimental analysis or discussion about the glucose-insulin control experiments. The discussion provided there was overly general (such that it should have been included in Section 3) and didn't really drive home expected insights or justifications for the claims made at the outset of the paper. There is some reference to Section K in the appendix for more detail but that was only to describe the experimental setup. There was no discussion about the results or additional analyses.

---

> ### Author Response · Authors · 2023-12-19
> **Thank you for the review**
>
> We thank the reviewer for the insightful comments and suggestions. Let us address them in order.
>
> 1. ``The setting and practical usage of the proposed IBNN is not made clear. It comes across as though the authors are trying to hide that the IBNN is merely a non-condensed ensemble of BNN models.''
>
> We agree with the fact that the presentation can and should be improved. In the updated version, we will make it clear that our procedure can be seen either as a BNN trained using credal prior and likelihood sets, or as an ensemble of uncountably infinitely many BNNs, that are not averaged out eventually. Rather, the posterior distributions that they ``generate'' are kept separate and form a (posterior) credal set. Such an infinite ensemble, though, is carried out using finitely many elements, that is, the extreme elements of the prior and likelihood credal sets. Here lies the appeal of our procedure.
>
>
> 2. ``On this note, it is unclear what the proposed IBNN is being presented for. Also, it seems that the authors have only focused on the BNN literature for uncertainty quantification.''
>
> Our procedure is introduced as an alternative to EBNN that (i) is better at disentangling aleatoric and epistemic uncertainties (AU and EU, resp.), and (ii) works better in safety-critical downstream tasks. In addition, credal sets are a step in the direction of answering a pivotal question in Bayesian analysis, that is, what are the "correct" prior and likelihood to use in a study. This is because the designer chooses sets of "plausible'' priors and likelihoods instead of just one of each. The set will be larger the more the ambiguity they face. We validate both claims (i) and (ii) with experimental evidence (Appendix I will be moved to the main body). Motivation (i) is important because designers may use our credal approach in procedures that return the "amount of'' AU and EU, and behave appropriately as a response. For example, in the presence of high AU, they may abstain from producing outcomes and query for human help, while in the presence of high EU, the model could be retrained with an augmented training set.  Being these the main goals, comparing our method with non-Bayesian techniques like the evidential-theory-based methodologies (Sensoy, Charpentier, Amini, etc.) or conformal prediction (Vovk, Shafer, Candès, Ramdas, etc.) needs a separate treatment, and is beyond the current scope. We will delve into it in future work. Instead, we will include a comparison between a BMA/Empirical Bayes' approach with our ``pessimistic'' one based on the lower probability, that is, the lower envelope of the predictive credal set.
>
> We note in passing that EU cannot be gauged using a single BNN (as pointed out by Hüllermeier and Waegeman). This because selecting a unique prior and a unique likelihood implicitly assumes perfect knowledge around the true prior and the true data generating process. In turn, a unique posterior is only able to retrieve AU. Methods that disentangle between the two types of uncertainties that are based on a single distribution are ad-hoc, but are not theoretically well-justified.
>
>
> 3. ``If the major claims of the paper are that by using IP to better ensemble BNNs, then it should be written that way.''
>
>
> We agree with the perception that the description of how the procedure is implemented, and of how predictions are made, are not sufficiently detailed. This is due to our attempt of keeping the paper to a reasonable length. Of course, though, this should not and cannot happen at the expense of clarity. Hence, in the updated version, we will deliver the implementation details in the most transparent fashion possible. We will also restructure section 4 to improve the way it is articulated, and we will be more direct about how the IHDR's are computed in practice. In particular, single HDRs can be routinely implemented, e.g., in $\mathsf{R}$, using package $\mathsf{HDInterval}$.
>
>
>
> 4. ``The additional experiments contained in Section 5 are woefully incomplete and have some shifted emphasis away from the stated advantages and claims of the paper.''
>
>
> We understand the reviewer's concerns about the analysis of the experiments on downstream tasks. In the revised version, we will deepen and further our analysis of the experiments -- and in particular of the one concerning the glucose-insulin dynamics -- to shed light on why and how our approach improves over EBNN.  The main idea being better uncertainty quantification leads to positive impact on the decisions made using them. We also plan to bring the comparison of epistemic and aleatoric uncertainty measurements to the forefront of the paper, to be more inline with the promises in the previous sections.

---

> ### Comment · Reviewer_tXse · 2023-12-21
> **Cannot move forward without a full revision**
>
> Thank you for your response, several of your points have helped clarify my stance on the paper. I am however unable to fully validate the statements made and the suggested improvements without a full revision of the paper to review. I will not address further discussion until I can verify that the proposed changes have been made since the paper, as submitted, requires a pretty substantial re-write.

---

### Review · Reviewer_cXJk · 2023-11-14

**Summary Of Contributions:**

The paper introduces Imprecise Bayesian Neural Networks (IBNNs), which allow for taking into account multiple priors and likelihoods and offer a different means of quantifying aleatoric and epistemic uncertainty. In empirical studies, they compare IBNNs to an ensemble of BNNs for both regression and classification (in the appendix).

The paper suggests using multiple likelihoods and priors and simply performing Bayesian inference on all combinations to obtain a set of BNNs.

To quantify and disentangle aleatoric and epistemic uncertainty for an input the paper suggests determining the highest $\overline{H}(P)$ and lowest entropy $\underline{H}(P)$ and then using the decomposition:

$$\underbrace{\bar{H}(P)}\_{\text {total uncertainty }}=\underbrace{\underline{H}(P)}\_{\text {aleatoric uncertainty }}+\underbrace{[\bar{H}(P)-\underline{H}(P)]}\_{\text {epistemic uncertainty }}.$$

To make predictions, the paper defines regions of imprecise high density and when it has to make a single prediction, it uses the mean of all predictions (I think?)

**Audience:**

Yes

**Claims And Evidence:**

No

**Requested Changes:**

Given the above, for acceptance, it would be great :

1. to expand the related work to cover comparison to uncertainty quantification using BNNs and compare to meta-priors,
2. to add comparisons to single BNNs as a baseline using the mean, stddev output parameterization as the ensembling for EBNNs might not be necessary,
2. compare to BMA as the cited paper is specific to HMC-based inference, which you do not seem to use anyway,
2. compare more explicitly to existing decompositions of aleatoric and epistemic uncertainty,
2. fix/improve/clarify the points about the comments on Prop 7 and the IHDR I have mentioned above, and
2. fix a few typos and hanging sentences and small issues throughout the paper:
### Typos & Misc

In the abstract:
1. “These latter are trained […] IBNNs” ← the former (BNNs), whereas the latter (IBNNs)
2. “We show that IBNNs performs better when  compared to an ensemble of BNNs **benchmark**” benchmark?
3. The procedure in 3.1 does not specify what happens after the posteriors are computed. There should be a step that explains how the IBNNs are evaluated because it does not make sense to me otherwise. (Esp given that the two steps are exactly how I would train an ensemble of diverse BNNs as well.)
4. Figure 7: what are the red boxes? this is not explained in the caption.

**Strengths And Weaknesses:**

The paper addresses an important point that often comes up with Bayesian neural networks and Bayesian approaches in general: the question of selecting a prior and likelihood to perform Bayesian inference on. Indeed, an often-heard claim against BNNs is that the choice of prior and likelihood can appear arbitrary, and while Bayesian inference is principled and rational, the Bayesian viewpoint does not help much with the former. It provides a good overview and many references for underlying research on uncertainty quantification and Bayesian theory.

Thus, the research question is of great importance and *the paper will find interest within the community*.

At the same time, there are a few weaknesses:

1. The paper mentions Kendall & Gal and other works that explicitly disentangle epistemic and aleatoric uncertainty with a single BNN but then claims that BNNs cannot disentangle aleatoric and epistemic uncertainty. Thus, related work that is mentioned in this context is not compared or acknowledged properly.
2. On the face of it, the approach trains an ensemble of BNNs with different architectures and/or priors (see also Wenzel et al., 2020). As such, it would seem necessary to compare to approaches that use meta-priors as this could be viewed as Bayesian inference over meta-priors where we choose a uniform (uninformative) prior over the prior distributions and likelihoods.
3. The experiments only compare an ensemble of BNNs with the proposed method. While the paper mentions that they do not apply BMA (Bayesian model averaging) due to a paper that found that it does not well for BNNs approximated with certain settings for HMC, a comparison to regular BNNs (not ensembled) and different approximation methods would have been helpful to compare the quality of the proposed method.

Thus, I don’t see all the statements as sufficiently evidenced yet.

## Details

The main difference between an ensemble of BNNs (EBNN) and the proposed imprecise BNN (IBNN) is that for the former the paper uses the variance of individual ensemble members (BNNs) as aleatoric uncertainty and the different to the total variance (ie the variance of the prediction means of each ensemble member via the law of total variance) as epistemic uncertainty, and for IBNNs it uses above decomposition between the highest and lowest-uncertainty member. There does not seem to be any difference in training etc.

The problem with the comparison between EBNNs and IBNNs is that for EBNNs, there is not necessarily a clean disentanglement because each BNN might also capture some of the epistemic uncertainty within itself. That is, a sufficiently powerful model class could learn all possible uncertainty itself such that the resulting ensemble would express no epistemic uncertainty whatsoever.

In contrast to this, the original deep ensemble paper (Lakshminarayanan et al., 2016) had each model output a mean and variance (or standard deviation) of an output Normal distribution explicitly, which limits what the model can express and makes it easier for it to constitute aleatoric uncertainty. One could thus create a BNN using such a model and then capture epistemic uncertainty as difference between the total variance vs an individual parameter draw’s mean and variance without the need to train multiple BNNs (as an EBNN).

It would be important to compare how such a single, simple BNN compares to IBNNs.

At the same time, this model could also be expanded to EBNNs by measuring epistemic uncertainty across all models while computing aleatoric uncertainty via the mean std. dev. outputs.

## On the theory

All the propositions in the main paper seem very simple and immediately follow from the properties of convex functions and sets (but also potentially weaker than they could be):

Proposition 2 follows from having a linear function on a convex set.
Proposition 4 holds not just for metrics and divergences but any function $f$ because it essentially only applies the definition of the infimum. You always have $\inf\_{x \in X} f(x) ≤ f(x’)$ for all $x' \in X$.
Proposition 7 follows from $\sup A \le \sup B$  for $A \subseteq B$.

Further:
> An immediate consequence of Proposition 7 is that the lower entropy of the extreme elements in $\Pi$ is an upper bound for the lower entropy of the whole credal set $\Pi^{\prime}$.

which follows from $\inf A \ge \inf B$  for $A \subseteq B$ is too weak and can be made stronger: entropy is a concave function. Hence:
$$
\inf\_{P \in \Pi} H(P) \le \sum \alpha\_j H(P\_j) \le H(\sum \alpha\_j P\_j),
$$
where $\sum \alpha\_j P\_j = \inf\_{P' \in \Pi’} H(P')$  and $P\_j \in \Pi$ etc, which follows from Jensen’s inequality. Hence, together with the definition of $\inf$, you actually have equality and now only an upper-bound.

> Finally, compute the $\alpha$-level Imprecise Highest Density Region (IHDR) $IR\_\alpha:=\cup\_{k=1}^N R\left(\hat{p}\_k^\alpha\right)$. By taking the union of the HDR's, we ensure that all the probability measures in the predictive credal set Conv $\hat{\mathcal{P}}$ assign probability of at least $1-\alpha$ to the event $\{y \in IR\_\alpha\}$; this is a consequence of Lemma 2. In turn, this implies that $\underline{\hat{P}}\left(y \in IR\_\alpha\right)=\inf\_{k \in\{1, \ldots, N\}} \hat{P}\_k\left(y \in IR\_\alpha\right) \geq 1-\alpha$.

I think this might be wrong and the IHDR should be defined as an intersection
$$IR_\alpha:=\bigcap_{k=1}^N R\left(\hat{p}_k^\alpha\right)$$
if you want to have the property you describe below. Otherwise, it might well be that the  $1-\alpha$ inequality does not hold for all $y$ for all $\hat{P}_k$?

### Meta-priors

We can introduce a random variable $K$ that denotes the prior $\times$ likelihood combination. Then we can set up the following probabilistic model:
$$
p(y|x, \theta, k) = p(y|x, \theta, k) \, p(\theta|k) \, p(k),
$$
if we choose a uniform prior we recover the approach introduced by the paper:
$$
\begin{aligned}
p(y|x,\mathcal{D}) &= \mathbb{E}\_{p(k)} \, \mathbb{E}\_{p(\theta | \mathcal{D}, k)} p(y|x, \theta, k) \\
&\approx \mathbb{E}\_{p(k)} \, \mathbb{E}\_{q_k(\theta)} p(y|x, \theta, k),
\end{aligned}
$$
where we approximate $p(\theta | \mathcal{D}, k)$ using $q\_k(\theta)$ via variational approximation:
$$
q\_k(\theta) := \arg\min\_q \,\, D\_{KL}(q(\theta) || p(\theta | \mathcal{D}, k)).
$$
In this context, it would be interest to compare to Empirical Bayes and use a more informative prior for $p(k)$.

### Aleatoric and Epistemic Uncertainty

If we take the view that IBNNs are using uninformative meta-priors, then we have we can compare them more directly with the definitions of aleatoric and epistemic uncertainty as used by Kendal & Gal and Smith & Gal, where they are defined as average/conditional entropy and mutual information as different between the BMA entropy and average/conditional entropy.

Concretely, the definition of aleatoric uncertainty in that view is:
$$H(P_K | K) = \frac{1}{N} \sum\_i H(P\_i)$$
and the definition of epistemic uncertainty is:
$$
I(P_K ; K) = H(\frac{1}{N} \sum\_i P\_i) - \frac{1}{N} \sum_i H(P\_i) \ge 0.
$$
Let’s compare this to your definitions.
$$
\begin{aligned}
\overline{H}(P) \ge H(\frac{1}{N} \sum\_i P\_i) &\ge H(P\_K|K) \ge \underline{H}(P) \text{ and thus}\\\\
\underline{H}(P) \le H(P\_K|K) \quad &\land \quad \overline{H}(P) - \underline{H}(P) >= I(P\_K;K).
\end{aligned}
$$
Thus, the proposed quantities are “wider and more extreme” than the ones introduced for BNNs conventionally.

**But** it would be great to compare their performance more directly.

A potential weakness of IBNNs is that if some of the priors or likelihoods is very bad, they might lock in both $\overline{H}(P)$ and $\underline{H}(P)$. For example, if one of the models severely underfits, $\overline{H}(P)$ might always be maximal while if one of the models severely overfits, $\underline{H}(P)$ might always be 0. Given that we can choose the likelihoods freely, one could easily create likelihoods that barely update a model or totally overfit (by taking the likelihood to a power for example).

---

Bibliography:

Wenzel, Florian, Jasper Snoek, Dustin Tran, and Rodolphe Jenatton. "Hyperparameter ensembles for robustness and uncertainty quantification." _Advances in Neural Information Processing Systems_ 33 (2020): 6514-6527.

Lakshminarayanan, Balaji, Alexander Pritzel, and Charles Blundell. "Simple and scalable predictive uncertainty estimation using deep ensembles. arxiv e-prints, page." _arXiv preprint arXiv:1612.01474_ 5 (2016).

---

> ### Author Response · Authors · 2023-12-19
> **Thank you for the review**
>
> We thank the reviewer for the insightful comments and suggestions. Let us address them in order. We mention in passing that, when our procedure has to make a single prediction, it considers the prediction that has the highest lower probability.
>
> First, we are particularly grateful about the insight on the original deep ensemble paper, whose methodology we will use as a baseline in the updated version.
>
> ***ON THE THEORY***
>
> Let us now address the theory concerns. We agree about the remarks on Propositions 2 and 4. We decided to name them Propositions -- as opposed to Theorems -- precisely because deriving them was relatively easy. We will add the suggested remarks, so that the interested reader can appreciate the more general results that can be immediately derived from them. Regarding the consequence of Proposition 7, if we understand correctly, the proof goes as follows. We are grateful for the suggestion.
>
> ***Proof*** Let $\Pi=\{P_1,\ldots,P_k\}$ and $\Pi^\prime=\mathrm{Conv}\Pi$. Pick any $P^\prime \in \Pi^\prime$. By the definition of $\Pi^\prime$, there exists a collection of non-negative reals $\beta_j$, $j=1,\ldots,k$, such that $\sum_{j=1}^k \beta_j=1$ and $\sum_{j=1}^k \beta_j P_j=P^\prime$. By the concavity of the entropy, we have that
>     $$H(P^\prime)=H\left( \sum_{j=1}^k \beta_j P_j \right) \geq \sum_{j=1}^k \beta_j H(P_j) \geq \sum_{j=1}^k \beta_j \underline{H}(P)=\underline{H}(P) := \inf_{P\in\Pi} H(P).$$
> Since $P^\prime$ was chosen arbitrarily and $\Pi$ is finite, this implies that
> $$\inf_{P^\prime\in\Pi^\prime} H(P^\prime)=:\underline{H}(P^\prime) \geq \underline{H}(P).$$
> In addition, we have that, since $\Pi \subset \Pi^\prime$, $\underline{H}(P^\prime) \leq \underline{H}(P)$. Combining this with the above result, we obtain
> $$\underline{H}(P^\prime) = \underline{H}(P).$$ ***End Proof***
>
> Finally, the reviewer's intuition about IHDRs seems to be wrong. Think of the following simple example. Consider three univariate Normals having the same variance but different means. Their HDRs are intervals around the means, of the form $(\mu-\epsilon,\mu+\epsilon)$. If the means are sufficiently apart from each other, then the intersection of the HDRs is empty. Instead, by taking the union, we have
> $$IHDR=\cup_{k=1}^3 HDR_k=\lbrace{y\in\mathcal{Y}: p_k(y) \geq \min_{k\in\lbrace{1,2,3\rbrace}} p_k^\alpha\rbrace},$$
> for some $\alpha\in (0,1)$, where $p_k(\cdot)$ is the pdf of the $k$-th Normal, and $p_k^\alpha$ is the largest constant such that $P_k [ y \in HDR_k ] \geq 1-\alpha$. Now, pick any $\tilde{P} \in \mathrm{Conv}(P_1,P_2,P_3)$. Then,
> $$\tilde{P}[y\in IHDR]= \sum_{k=1}^3 \gamma_k P_k [y\in IHDR] \geq \sum_{k=1}^3 \gamma_k P_k [ y \in HDR_k ] \geq 1-\alpha,$$
> where $\lbrace{ \gamma_k \rbrace}$ is a collection of non-negative reals such that $\sum_{k=1}^3 \gamma_k=1$ and $\sum_{k=1}^3 \gamma_k P_k=\tilde{P}$. The first inequality comes from the fact that $\mathrm{IHDR}=\cup_{k=1}^3 \mathrm{HDR}_k$.

---

> > ### Author Response · Authors · 2023-12-19
> > **Continuation**
> >
> > ***META PRIORS***
> >
> > The process suggested by the reviewer can be used to select one distribution from the credal predictive set. While very interesting in its own right, this is not what our method does. In fact, it would actually defy the purpose of working with credal sets. During training, our method produces a set of posterior distributions, which, during inference, is used to derive a set of predictive distributions. They are kept separate; we do not average over them. The output at inference time, then, is either the predictive IHDR, or the prediction having the highest lower probability. We will make this clear in the new version of our manuscript.
> >
> > Let us also add that comparing a BMA/Empirical Bayes' approach with our ``pessimistic'' one based on the lower probability can be of interest, and will be included in the updated version of the paper.
> >
> > ***ALEATORIC AND EPISTEMIC UNCERTAINTIES***
> >
> > In light of what we pointed out in the previous subsection, it is unsurprising that the measures for aleatoric and epistemic uncertainties used by Kendal \& Gal and Smith \& Gal are ``narrower and less extreme'' than the ones for our procedure. The reason being that they are based on only one distribution, while our measures are based on credal sets, i.e. on uncountably infinitely many distributions. In the updated version of the paper, we will make this clearer, and introduce a comparison between these measures, similarly to what we did for EBNNs measures.
> >
> > Also, while we agree with the reviewer that a "bad choice'' of priors and likelihoods may lead to maximal upper and lower entropy, this is not a risk confined to our procedure. Poor modeling choices are an unavoidable risk in model-based techniques. This gave rise to the famous adage by Box and Draper "essentially, all models are wrong but some are useful''. We maintain that our method is indeed useful, since it overcomes some of the shortcomings of traditional Bayesian techniques -- as explained in Appendices A and B. As for "regular'' Bayesian methods, though, for our approach too the designer will need to make ``plausible'' choices for priors and likelihoods. We will be more explicit about this in the updated version.
> >
> > ***REQUESTED CHANGES***
> >
> > Let us recap what we plan to do to address the reviewer's comments.
> >
> > (i) Expand the Related Work section as suggested;
> >
> > (ii) Add the desired comparisons with BMA/Empirical Bayes and the original deep ensemble paper;
> >
> > (iii) Compare our AU/EU decomposition's with that of Kendal \& Gal and Smith \& Gal;
> >
> > (iv) Improve Proposition 7 and clarify the definition of IHDR;
> >
> > (v) Fix the noted typos. We will also make it clearer in the Procedure in section 3.1 that the posteriors are used -- during inference -- to derive the predictive distributions, which in turn are used to produce an IHDR or to find the output having the highest lower probability.

---

> > > ### Comment · Reviewer_cXJk · 2023-12-27
> > > **Thank you for your comments**
> > >
> > > Thank you for answering my questions and offering changes and clarifications to the submission accordingly.
> > >
> > > I concur with the other reviewer that I would like to see the changes as there seem to be quite a few edits overall.
> > >
> > > If you and the action editor agree, I would suggest that we extend the rebuttal period accordingly, so you can incorporate the changes into the paper, and we can have a look before accepting the paper. This might be easier than asking for a major revision.
> > >
> > > Thanks and best wishes. (And happy holidays!)

---

> > > > ### Author Response · Authors · 2023-12-27
> > > > **Thank you!**
> > > >
> > > > Dear Reviewer cXJk,
> > > >
> > > > We are doubly grateful, first for having spotted the inaccuracy in our explanation, and second for having proposed to extend the rebuttal period. We of course accept, as we are working hard on the new version of the manuscript!
> > > >
> > > > Once again we are very grateful, and happy holidays to you as well!
> > > >
> > > > Yours,
> > > >
> > > > Authors

---

> > ### Comment · Reviewer_cXJk · 2023-12-27
> > **Re: IHDR argument**
> >
> > Thank you for your reply. I see your argument. While
> >
> > > $$IR_\alpha:=\cup_{k=1}^3 HDR_k=\\{ y \in \mathcal{Y}: p_k(y) \geq \min_{k \in\{1,2,3\}} p_k^\alpha \\}$$
> >
> > seems incorrect as the union is just $\\{ y \in \mathcal{Y}: \exists k: p_k(y) \geq p_k^\alpha \\}$ where $k$ has to be the same for both LHS and RHS, the union ensures that $\hat{P}\_k\left[y \in IR\_\alpha\right] \geq \hat{P}\_k\left[y \in R\left(\hat{p}\_k^\alpha\right)\right] \geq 1-\alpha$ and the inequality chain holds for any convex combination as well. Apologies for this.

---

### Review · Reviewer_LjWU · 2023-12-06

**Summary Of Contributions:**

The paper presents Imprecise Bayesian Neural Networks, which extend Bayesian Neural Networks to credal sets. The IBNN is introduced. A series of theoretical results is presented. Practical aspects are discussed, followed by experimental results. The paper ends with a discussion of related work and concludes by summarizing the contributions.

**Audience:**

No

**Broader Impact Concerns:**

None.

**Claims And Evidence:**

No

**Requested Changes:**

It is my opinion that the paper needs to be rewritten.

**Strengths And Weaknesses:**

I am, unfortunately, quite confused by the paper. My confusion begins with the introduction of the paper, which is quite diffuse. What is the goal of the paper? Going to the conclusion, one might infer that it is to extend the BNN, however, there is not a clear statement of that problem and why it is interesting or challenging. (There is extended discussion of other things, but it does not focus on BNNs.) In the Background and Preliminaries, it is not clear to me where the BNN is defined? In addition, the math is not clearly introduced, making it more difficult to understand what the authors intend (see below for further discussion). It is an unusual choice for the first mathematical statement to be a remark. What are we remarking on? I would have expected to see a definition of IBNNs or a numbered equation at least. I am not really sure what they are. Definition 3, which is in the theory section, appears intended to define the IBNN, however, I do not recognize it as a mathematical statement. From this point on, I was unable to follow the paper. What follows is further comments that supply a few more details on questions that arose when reading.

Comments:
I find the introduction to be far too broad to appropriately contextualize the contribution of the current work. What is the *new* contribution of this work? The references to old philosophical debates are a distraction from the point of this paper. It seems like the cleanest framing of the paper is to extend BNNs to allow imprecise priors. If so, what are the technical challenges that are associated with doing that? I assume no one has done this before (though the introduction is not focused enough to give me confidence). If so, why not?

The mathematical introduction is strangely pseudo formal. What are H and D? They are stated as hypothesis and data, but from what set are they? Does it matter? Conventionally, I would assume the set be capitalized and the elements not? (Of course, conventions can differ, but I can't place the notational style, which makes reading difficult.) In the first paragraph of 2.1 P is capitalized for probability, whereas it is lowercase in the next? Is D_x \times D_y the training set? That would suggest that every input is paired with every output? What is a generic space?


Minor comments:
- " They allow to distinguish between aleatoric and epistemic uncertainties, and
to quantify them." I would strongly recommend replacing pronouns like they/them with the referents. Replacing pronouns improves readability by reducing the load on the reader.
- " cannot be directly used on out-of-distribution data without further examples" why not?
- "it overcomessome of the drawbacks of deep learning models, namely that they are prone to overfitting, which adversely affects their generalization capabilities" Really?
- "This is desirable in light of several areas of recent ML research, such as Bayesian deep learning (Depeweg et al., 2018; Kendall & Gal, 2017), adversarial example detection (Smith & Gal, 2018), and data augmentation in Bayesian classification (Kapoor et al., 2022)." This sentence does not tell me why this is desirable.
- It is a very unusual choice to refer the reader to the Appendix in the introduction! Arguably the point of an introduction is to introduce. If it doesn't do that, what is it doing?

---

> ### Author Response · Authors · 2023-12-19
> **Thank you for the review**
>
> We thank the reviewer for the insightful comments and suggestions. Let us
> address them in order.
>
> 1. ``What is the goal of the paper?''
>
> This question is similar to that of Reviewer tXse. Our procedure is introduced as an alternative to EBNN that (i) is better at disentangling aleatoric and epistemic uncertainties (AU and EU, resp.), and (ii) works better in safety-critical downstream tasks. In addition, credal sets are a step in the direction of answering a pivotal question in Bayesian analysis, that is, what are the "correct'' prior and likelihood to use in a study. This because the designer chooses sets of "plausible'' priors and likelihoods instead of just one of each. The set will be larger the more the ambiguity they face. We validate both claims (i) and (ii) with experimental evidence (Appendix I will be moved to the main body). Motivation (i) is important because designers may use our credal approach in procedures that return the ``amount of'' AU and EU, and behave appropriately as a response. For example, in the presence of high AU, they may abstain from producing outcomes and query for human help, while in the presence of high EU, the model could be retrained with an augmented training set.
>
> 2. ``In the Background and Preliminaries, it is not clear to me where the BNN is defined?''
>
> BNNs are defined is Section 2.1.
>
> 3. ``It is an unusual choice for the first mathematical statement to be a remark. What are we remarking on?''
>
> Remark 1 is not a mathematical statement. It ***remarks*** the notation for $\Pi$ and $\Pi^\prime$ that we use in the paper. It also contains a quick definition of a finitely generated credal set, which is also depicted in Figure 1 to make it easier to understand.
>
> 4. ``I would have expected to see a definition of IBNNs or a numbered equation at least.''
>
> In the updated version of the manuscript, the definition of our procedure will be put in section 3.1 in the form of an algorithm. During training, our method derives the credal posterior set, which is used, during inference, to obtain a credal predictive set. The latter is used to form an IHDR, or to find the prediction having the highest lower probability.
>
> 5. ``Definition 3, which is in the theory section, appears intended to define the IBNN, however, I do not recognize it as a mathematical statement.''
>
> See the answer to the previous comment. We note in passing how a definition is not a mathematical statement -- like Propositions, Lemmas, or Theorems -- but rather, a way of introducing an object.
>
> 6. ``From this point on, I was unable to follow the paper. What follows is further comments that supply a few more details on questions that arose when reading.''
>
> We are sorry the reviewer was not able to follow the rest of the paper, and we appreciate their herculean effort to give us feedback.
>
> 7. ``I find the introduction to be far too broad to appropriately contextualize the contribution of the current work. What is the new contribution of this work?''
>
> The answer to this comment is very similar to that of comment 1.
>
> 8. ``The references to old philosophical debates are a distraction from the point of this paper. It seems like the cleanest framing of the paper is to extend BNNs to allow imprecise priors. If so, what are the technical challenges that are associated with doing that? I assume no one has done this before (though the introduction is not focused enough to give me confidence). If so, why not?''
>
>
> Being this a journal paper, we feel the references to the works that started the field of imprecise probabilities are not only in order, but dutiful, and hardly a distraction from the main motivation on why our method is introduced. They are needed to motivate how the type of credal sets that we use in the paper -- namely, finitely generated credal sets -- came about. Specifying a unique prior and likelihood is often times impossible, or it is done for convenience. At the same time, taking the frequentist route of (heuristically) considering ***all possible priors*** may be inefficient. Eliciting a set of plausible priors and likelihoods, instead, captures the idea of being aware of the existence of a true prior and likelihood, but only being able to specify sets to which these ideal distributions belong. Of course, when we say this, we must cite who introduced these concepts first. The remaining questions were answered in question 1.

---

> ### Author Response · Authors · 2023-12-19
> **Continuation**
>
> 9. ``The mathematical introduction is strangely pseudo formal. What are $H$ and $D$? They are stated as hypothesis and data, but from what set are they? Does it matter? Conventionally, I would assume the set be capitalized and the elements not? (Of course, conventions can differ, but I can't place the notational style, which makes reading difficult.) In the first paragraph of 2.1 P is capitalized for probability, whereas it is lowercase in the next? Is $D_x \times D_y$ the training set? That would suggest that every input is paired with every output? What is a generic space?''
>
>
>
> While we welcome their feedback, we disagree with the reviewer. Our introduction is not ``strangely pseudo-formal''. Rather, it uses the adequate degree of formality needed for a paper that is mainly of theoretical nature. As explained in section 2.1, $H$ depicts a hypothesis, and $D$ the available data. They are both sets. Since in section 2.1 we work with the highest level of generality, we do not assume that the hypotheses/parameter space $\Theta$ is finite or countable. As a consequence, $H$ must be a (hypotheses) set, $H\subset \Theta$. Notice here that a hypothesis is of the form $H=\lbrace{\theta\in A\rbrace}$, where $\theta$ is the parameter of interest, and $A$ is a subset of the parameter space $\Theta$. Also, $D$ is a set since, for example, $D=\lbrace{(x_i,y_i)\rbrace}$, $i=1,\ldots,n$, where $x_i$ is the $i$-th observed input, and $y_i$ is the $i$-th observed output. In addition, the difference between $P$ and $p$ is the classic one between probability ***measures*** and probability ***density or mass functions***. Yes, $D=D_\mathbf{x} \times D_\mathbf{y}$ is the training set, as pointed out in (a), page 3. We have that $D_\mathbf{x}=\lbrace{x_i\rbrace}$ and $D_\mathbf{y}=\lbrace{y_i\rbrace}$, $i=1,\ldots,n$. Notice that the subscripts for both $D_\mathbf{x}$ and $D_\mathbf{y}$ are in bold face so that they do not get confused with the elements of the sets. As it is often the case, in a training set we have some input $x_i\in D_\mathbf{x}$ that is paired with its correct output $y_i\in D_\mathbf{y}$. A generic space is a space/set with no assumed mathematical structure (e.g. it is ***not*** assumed that is normed, Hilbert, Polish, etc.).
>
> 10. $\ll$" They allow to distinguish between aleatoric and epistemic uncertainties, and to quantify them." I would strongly recommend replacing pronouns like they/them with the referents. Replacing pronouns improves readability by reducing the load on the reader.$\gg$
>
> While we agree with the reviewer, in the noted sentence it is extremely clear that ``them'' refers to epistemic and aleatoric uncertainties, that were just then mentioned.
>
> 11. $\ll$" cannot be directly used on out-of-distribution data without further examples" why not?$\gg$
>
>
> Because further examples are needed to constitute the validation set. This too was clear from the whole sentence, which reads as ``While such methods are a promising first step, they require a calibration set (in addition to the original training set) and cannot be directly used on out-of-distribution data without further examples.''
>
> 12. $\ll$"it overcomes some of the drawbacks of deep learning models, namely that they are prone to overfitting, which adversely affects their generalization capabilities" Really?$\gg$
>
> Yes, this was studied by Jospin et al. in their 2020 paper ``Hands-on Bayesian Neural Networks – A Tutorial
> for Deep Learning Users'', as we cite in our work.
>
> 13. $\ll$"This is desirable in light of several areas of recent ML research, such as Bayesian deep learning (Depeweg et al., 2018; Kendall \& Gal, 2017), adversarial example detection (Smith \& Gal, 2018), and data augmentation in Bayesian classification (Kapoor et al., 2022)." This sentence does not tell me why this is desirable.$\gg$
>
>
> The sentence means that quantifying and disentangling AU and EU has been deemed desirable by the cited papers, that accrue to different fields of ML/AI, namely Bayesian deep learning, adversarial example detection, and data augmentation in Bayesian classification. We feel the sentence is clear enough, as similar sentence constructions are common, see e.g. Sale et al., 2023, ``Is the volume a good measure for epistemic uncertainty?''

---

> > ### Author Response · Authors · 2023-12-19
> > **Continuation**
> >
> > 14. ``It is a very unusual choice to refer the reader to the Appendix in the introduction! Arguably the point of an introduction is to introduce. If it doesn't do that, what is it doing?''
> >
> >
> > Here the reviewer confuses us. First they say they don't want references to ``old philosophical papers'' in the introduction, then when we refer to more philosophical motivations in the Appendices, they complain. The Appendices are referred to in the Introduction -- and also throughout the manuscript -- whenever their content is relevant to the ideas we are presenting. Because we reckon that they are not crucial to the main part of the paper, we relegate them to the appendices. Nevertheless, the interested reader must be referred to them, so that they can delve further into details.
> >
> >
> > 15. ``It is my opinion that the paper needs to be rewritten.''
> >
> >
> > While we treasure the reviewer's opinion, we strongly feels it is unjustified, given also that -- by their own admission -- they stop reading after Definition 3. That is, after page 4 out of 35. Nonetheless, we will put an effort to make the paper easier to read.

---

### Author Response · Authors · 2023-12-19
**Title change in new version**

In the updated version of our manuscript, we will change the title to "Credal Bayesian Deep Learning''. In our paper, the neural net component is a finite collection of BNNs (there are no architectural details), whereas our actual contribution is the imprecise/credal framework that encapsulates such BNNs. In the new version, we will dispense with Definition 3, and change the Procedure in 3.1 to an actual Algorithm with During Training/During Inference parts. During training we obtain the posterior credal set, and during inference we derive the predictive credal set, which is then used either to output the IHDR, or the prediction having the highest lower probability.

---

### Decision · Action_Editor_UrfG · 2024-01-11

**Recommendation:** Reject

**Comment:**

The authors engaged in a fruitful discussion with the reviewers promising to tackle or clarify most of the points raised during the review process in a revised version of the manuscript.
However, these changes are too substantial to be handled by a minor revision.
Given the relevance of the work and the concrete suggestions discussed, I encourage the authors to continue with their research, to revise their manuscript accordingly, and to resubmit it.

**Audience:**

Bayesian neural nets (BNNs) are an active area of research within the TMLR community. The same holds for uncertainty quantification, which is relevant throughout many application areas. The authors approach both tasks and are thus relevant for

**Claims And Evidence:**

In its current form, the submission does not provide evidence for all of the claims it states and additionally requires further improvements in its overall structure.

**Resubmission Of Major Revision:**

The authors may consider submitting a major revision at a later time.